# SPECS: FASTER TEST-TIME SCALING THROUGH SPECULATIVE DRAFTS AND DYNAMIC SWITCHING

## ABSTRACT

Scaling test-time compute has driven the recent advances in the reasoning capabilities of large language models (LLMs). However, increased compute often comes at the expense of higher user-facing latency, directly impacting user experience. Current test-time scaling methods primarily optimize for accuracy based on total compute resources (FLOPs), often overlooking latency constraints. To address this gap, we propose SPECS, a latency-aware test-time scaling method. SPECS builds upon beam search, which generates multiple reasoning traces for each step with a reasoning model, and selects one to continue from based on the scores from a dedicated reward model. Inspired by speculative decoding, SPECS uses a smaller, faster model to generate candidate traces efficiently, and evaluates these candidates with both the reasoning model and the reward model. We design novel strategies to select candidate drafts using these model evaluations, including reward-guided soft verification, and a dynamic switching mechanism to defer to the larger model on harder steps. Empirical results on MATH500, AMC23 and OlympiadBench datasets show that SPECS matches or surpasses the accuracy of beam search while reducing latency by up to ∼18%. Our theoretical analysis shows that our algorithm converges to the solution of a KL-regularized reinforcement learning objective as the beam width grows.

## 1 INTRODUCTION

Modern LLMs excel at multi-step reasoning, and scaling test-time compute has played a major role in achieving these reasoning capabilities by letting these models tackle harder problems with extra "thinking" compute resources (Cobbe et al., 2021; Wei et al., 2022; Beirami et al., 2024; Brown et al., 2024; Beeching et al., 2024; Qiu et al., 2024; OpenAI, 2024; DeepSeek-AI, 2025).

To date, test-time scaling methods have primarily optimized performance based on total compute, demonstrating improved downstream task performance with increased FLOPs or generated tokens (OpenAI, 2024; Snell et al., 2025). However, user experience often depends more directly on serving *latency*, especially in low-throughput scenarios like personalized interactions or applications serving few users (Patel et al., 2024; Agrawal et al., 2024). Autoregressive generation from transformer-based LLMs often operates in the regime where the latency is limited by memory loading rather than total FLOPs (Tiwari et al., 2025; Yuan et al., 2024b). Consequently, we study the critical research question:

*Can we design test-time scaling methods which optimize for latency with minimal accuracy decrease?*

The problem of improving the latency of autoregressive generation has received significant attention, and resulted in approaches such as speculative decoding (Leviathan et al., 2022; Chen et al., 2023). Speculative decoding uses a smaller, faster draft model to propose candidate tokens, which are validated in parallel by a larger target model, reducing the total memory loading time and hence the overall decoding latency.

In this work, we study how speculative drafts can be used to develop new test-time scaling methods which optimize for latency. Our investigation builds on the family of beam search or tree search (Mudgal et al., 2024; Beeching et al., 2024; Sun et al., 2024; Qiu et al., 2024; Snell et al., 2025) algorithms for test-time scaling. In each iteration multiple candidate reasoning steps (i.e., beams) are generated in

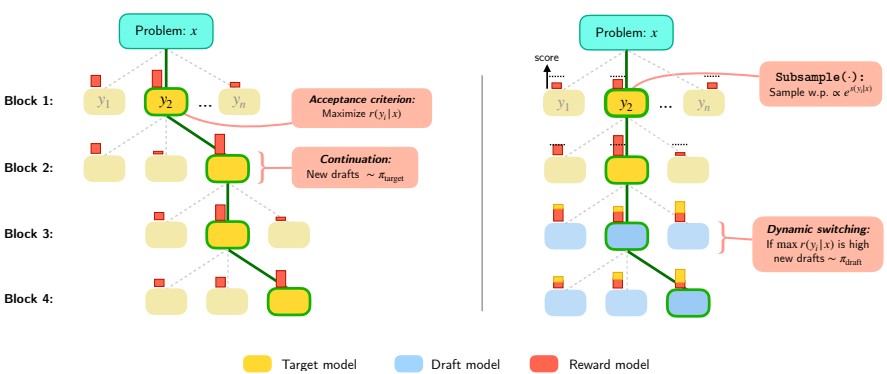

Figure 1: Visualization of Beam Search vs SPECS. In beam search the trajectories are generated by the target model ($p$) and scored using a PRM ($r$). In contrast, in SPECS the beams are dynamically switched to generation from the draft model, and scored by a combination of target and PRM model, resulting in better latency-performance tradeoff. Draft proposal and selection are further controlled by the SUBSAMPLE subroutine (see section 2 for details).

parallel from a target model, and the most promising ones are kept to continue generation from, based on a reward signal (i.e., a process reward model (PRM) (Zhang et al., 2025a) or self rewards (Yuan et al., 2024a)). The mutli-step structured exploration makes them particularly effective at balancing accuracy and computational effort. At the same time, it also offers the flexibility of dynamically switching reasoning models for easier/harder steps.

We propose a new algorithm, SPECS (SPECulative drafting for faster test-time Scaling). The algorithm relies on a speculative drafting step for improving latency. A speculative drafting step uses a fast draft model to propose candidate reasoning steps, and jointly uses a larger, more capable target model and a reasoning-specific PRM to select from the candidates. To make sure that our algorithm incurs minimal decrease in accuracy, we design novel dynamic switching scheme to identify reasoning steps where speculative drafting would not hurt latency, as well as a theoretically-grounded score metric integrating the evaluations of the large model and the reward model for better draft selection.

Our contributions can be summarized as follows:

1. We propose a novel test-time scaling algorithm, SPECS (Algorithm 1), which utilizes speculative drafts to reach favorable latency-accuracy tradeoff. Our algorithm starts by generating from the target model and dynamically switches to speculative drafting for high reward traces to minimize accuracy drop. This builds on the novel insight that the accuracy gap between a small model and a large model is small for these traces, which could be of independent interest. To the best of our knowledge, other works building on speculative drafting all start with a draft model and switch to target model when the reward signal is low, resulting in either lower accuracy or wasted latency for generating speculative drafts.

2. We design a novel strategy to integrate scores from target models and PRMs for better draft selection. Theoretically, we show that the approach converges gracefully to the optimal solution of a KL-regularized reward maximization objective as beam-width (i.e., parallel compute) increases, demonstrating the theoretical grounding of the proposed approach. We present a novel analysis of the soft best-of-$N$ algorithm, and extends this to speculative drafting and multi-step reasoning.

3. We evaluate on MATH-500, AMC23 and OlympiadBench datasets, demonstrating that SPECS achieves up to 18% reduction in latency while achieving on-par or higher accuracy compared to beam search and other baselines, using (Qwen-1.5B-Instruct & Qwen-7B-Instruct) and (Llama3.2-1B-Instruct & Llama3.1-8B-Instruct) as draft-target model pairs with Skywork-o1-Open-PRM-Qwen-2.5-7B as the process reward model.

We refer readers to appendix A for a detailed comparison to related work.

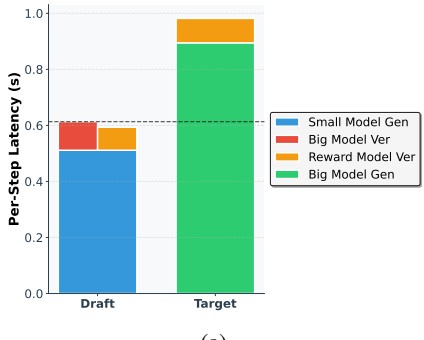 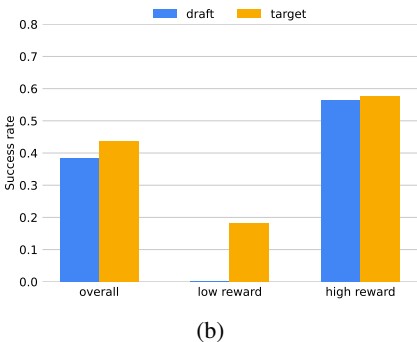

(a) (b)

Figure 2: **(a)** Latency of generation from the target model (Qwen2.5-7B-Instruct) vs. generation from the draft model (Qwen2.5-1.5B-Instruct) with scoring: we observe that latency savings from using the draft model to generate candidate blocks overcomes the overhead of scoring by the target model and PRM. **(b)** We generate the first $8$ steps of reasoning from the target model, and complete the remaining steps either using the draft model or the target model. The initial $8$-step partial reasoning traces generated by the target model are bucketed into high reward (PRM score at least $0.5$) and low reward (PRM score at most $0.5$). Using the draft model to complete high reward traces solves a very similar proportion of problems compared to if they were completed using the target model. The performance is dismal ($\approx 0$) when draft model is used to complete low reward traces.

## 2 SPECULATIVE DRAFTING WITH REWARD-GUIDED SOFT VERIFICATION

Before presenting the algorithm details, we discuss some general observations that lead to the development of our SPECS algorithm, which we believe could be of independent interest. The most costly step (in terms of latency) for beam search with a large target model is autoregressive generation to obtain the beams (i.e., candidate reasoning blocks). Our main technique for reducing this latency cost is to instead speculatively generate beams via autoregressive generation from a smaller draft model, which are then scored using the target model and reward model. Due to the fact that autoregressive sampling in modern LLMs is bottlenecked by memory bandwidth, evaluating a reasoning block with a target model (i.e., computing logits of each token) is much faster than generating a reasoning block from the same model, despite taking the same number of FLOPs. This also holds even when combined the draft model generation time, which is faster due to its smaller size. Moreover, the scoring by the target model and reward model can be executed in parallel to further reduce latency. While using small draft models for improving latency is the subject of several past work on speculative decoding (Leviathan et al., 2022), we carry out an experiment in Figure 2a to understand the typical per-step latency improvements with modern reasoning and reward models.

**Observation 1** (*Low-latency draft model generation*)**.** Generating multiple candidate sequences from the draft model, followed by scoring them using the larger target model and a reward model can be significantly faster compared to generating responses directly from the target model.

While generating reasoning blocks from a draft model is faster, their average quality will be worse since a less capable model is used for generation. Careful draft selection is needed to guarantee a minimal drop in accuracy. Speculative decoding (Leviathan et al., 2022; Sun et al., 2023; Miao et al., 2024) validates generated drafts by attempting to match the target model distribution. However, in the process, drafts may be unnecessarily discarded, even if they happened to collect high rewards. Thus, it is reasonable to expect that a soft verification mechanism, which relaxes the distribution matching guarantee can obtain a speedup, while still maintaining high accuracy on reasoning tasks.

On the other hand, reasoning datasets typically evaluate model capabilities across a range of skills and difficulty levels. At a finer granularity, certain steps of the solution to a problem might be easy, and a large target model may be overprovisioned to generate steps that the draft model can already solve. For harder steps, it makes sense to defer to the more capable target model for candidate generation.

These discussions motivate the following two questions,

1. How can we identify steps where speculative drafting will not significantly hurt accuracy?
2. Given a set of candidate steps how do we best select candidates based on scores from the target model and the reward model?

To gain intuition towards answering the first question, we study how the reward of reasoning trace evolves over the course of a generation, and how it relates to the final accuracy of the generated response. In fig. 2b we carry out two inference runs, where we start by generating 8 steps of reasoning from the target model. In one case, we continue the rest of the trace using the large target model, and in the other we switch to the draft model and continue thereon. When we condition on the traces having high reward (as computed by a PRM) at the 8th step, we observe that completing the trace using the draft model affects accuracy very minimally compared to completing with the target model. This observation motivates switching between the models based on process reward of the trajectory.

**Observation 2** (*Draft model can complete high-reward traces*)**.** Generating blocks from the draft model does not hurt accuracy when the PRM score of a partial trace is high.

As a counterpart to this observation, we note from Figure 2b that continuing generation from the draft model starting from low reward traces performs poorly. To minimize any decrease in accuracy from speculative drafting, it is reasonable to begin with generating steps from the target model until high reward traces are encountered.

## 2.1 THE SPECS ALGORITHM

With the insights developed above, we present our main algorithm SPECS. The algorithm follows an overall draft-and-select framework where the final response is generated block-by-block. In each iteration, the algorithm uses a generation model $\pi_{\text{gen}}$ to propose candidate blocks $\{Y_i\}_{i=1}^n$ to extend the current reasoning trace tr through autoregressive sampling. $\pi_{\text{gen}}$ is initialized to be a larger reasoning model $\pi_{\text{target}}$, and over the course of generation switches to a smaller draft model $\pi_{\text{draft}}$. When $\pi_{\text{gen}} = \pi_{\text{draft}}$, we call this a speculative drafting step.

The candidate blocks are then evaluated by both a larger, more capable reasoning model $\pi_{\text{target}}$ (for speculative drafting step), and a reward model $r$. A score (eq. (1)) is assigned to each candidate based on the evaluations, which is then used to select a candidate block (summarized in Algorithm 2). The selected block is appended to the existing response to begin the next iteration.

We use the score to adaptively decide which model to use to generate candidate blocks. This is most closely related to the approach of cascades (Dohan et al., 2022; Jitkrittum et al., 2023; Yue et al., 2024; Narasimhan et al., 2024) and aims to ensure that the larger model generates candidates for the harder steps of a problem, while the smaller draft model is used for the easier steps.

---

**Algorithm 1** SPECS meta algorithm

1: **Input:** Prompt $X$,
    Draft model $\pi_{\text{draft}}$,
    Target model $\pi_{\text{target}}$,
    Process reward model $r$,
2: **Hyperparameters:** Beam-width $n$,
      Block-size $\gamma$,
      Switching threshold $\tau$.
3: **Initialize:** Partial response tr $\leftarrow X$
4:    Draft generator $\pi_{\text{gen}} \leftarrow \pi_{\text{target}}$
    // Generation begins with target model.
5: **Repeat**
 *Step I.* Generate candidate blocks
6: Draw $n$ blocks $\sim \pi_{\text{gen}}$ of block-size $\gamma$,
$$\mathbf{Y}_{\text{draft}} = \{Y_i\}_{i=1}^n \overset{\text{i.i.d.}}{\sim} \pi_{\text{gen}}(\cdot|\text{tr})$$

 *Step II.* Select candidate block
7: $Y = \text{SUBSAMPLE}(\mathbf{Y}_{\text{draft}}, \beta, \pi_{\text{gen}}|\text{tr})$,
8: tr $\leftarrow (\text{tr}, Y)$   // SUBSAMPLE$(\cdot)$ selects a
        // draft using scores in eq. (1)

 *Step III.* Dynamic switching
9: **if** $\max_{i \in [n]} r(Y_i|\text{tr}) > \tau$
10:  $\pi_{\text{gen}} \leftarrow \pi_{\text{draft}}$
     // Switch to generating from draft
     // model if reward signal $r$ is large
11: **Until:** Target sequence length is achieved.
12: **Return:** tr

---

## 2.2 CANDIDATE SELECTION: SUBSAMPLE

In this section we describe the SUBSAMPLE mechanism which carries out the selection process of candidate blocks $\{Y_i\}_{i=1}^n$. The goal is to select a response which simultaneously has high reward (i.e., completions are likely to lead to a correct response) while not having very low probability under the larger target model. SUBSAMPLE will carry this out by combining the evaluations from both the large reasoning model and the re-

ward model in the following score function:

$$S_i = \log\left(\frac{\pi_{\text{target}}(Y_i|\text{tr})}{\pi_{\text{gen}}(Y_i|\text{tr})}\right) + \beta r(Y_i|\text{tr}) \quad (1)$$

When $\pi_{\text{gen}} = \pi_{\text{target}}$, this collapses to $r$. With $\pi_{\text{gen}} = \pi_{\text{draft}}$ this also includes the log density

---

**Algorithm 2** SUBSAMPLE subroutine

1: **Input:** Context tr, Draft blocks $\{Y_i\}_{i=1}^n$,
      Target model $\pi_{\text{target}}$, PRM $r$,
      Draft generator $\pi_{\text{gen}}$,
2: **Hyperparameters:** Threshold $\tau$, Temp. $\beta$
3: **Initialize:** $L \leftarrow [n]$
4: **for** $i = 1$ **to** $n$,
5:    Define,
   $$S_i = \log\left(\frac{\pi_{\text{target}}(Y_i|\text{tr})}{\pi_{\text{gen}}(Y_i|\text{tr})}\right) + \beta r(Y_i|\text{tr})$$

6: Sample $i^\star \in [n]$ with probability,
   $$P(i^\star = i) \propto e^{S_i}\mathbb{I}(i \in L)$$

7: **Return:** $Y_{i^\star}$

---

ratio of the block between draft and target models. We formalize the choice of score in Section 3.1 through the lens of optimizing the expected reward with KL regularization (to the target model).

### 2.3 DEFERRALS VIA DYNAMIC SWITCHING

SPECS uses a deferral rule to determine which (draft or target) model to use for generating candidate blocks in each subsequent iteration. Along the lines of Observation 2, the algorithm begins generation from the target model, since the PRM reward is low to begin with. The algorithm proposes to switch generating future reasoning steps from the draft model if high reward partial traces are present in the current set of beams (i.e., their reward exceeds a threshold). This builds on our finding in Observation 2: it is beneficial from a latency point-of-view to switch to generating from the draft model when we already have traces with high reward.

## 3 UNDERSTANDING SPECS FROM A THEORETICAL LENS

### 3.1 MOTIVATION BEHIND SCORES IN SUBSAMPLE

The design of the score (eq. (1)) in SUBSAMPLE is motivated by the objective of solving a one-step KL-regularized reward maximization problem (Jaques et al., 2019; Stiennon et al., 2020). For the base model $\pi_{\text{target}}$ and a reward function $r$, the objective is to maximize,

$$\mathcal{L}_\beta(\pi) = \mathbb{E}_{\rho,\pi}[r(Y|X)] - \frac{1}{\beta}D_{\text{KL}}(\pi\|\pi_{\text{target}}). \quad (2)$$

The optimal solution to this objective has an explicit form (Korbak et al., 2022b;a; Yang et al., 2024b):

$$\pi_\beta^\star(y|x) \propto \pi_{\text{target}}(y|x)e^{\beta r(y|x)}. \quad (3)$$

For a different generator model $\pi_{\text{gen}}$, let $S(y|x) = \log(\pi_{\text{target}}(y|x)/\pi_{\text{gen}}(y|x)) + \beta r(y|x)$. Then, the optimal policy can be written down in a different form by importance reweighting. Namely,

$$\pi_\beta^\star(y|x) \propto \pi_{\text{gen}}(y|x)e^{S(y|x)}. \quad (4)$$

Thus, at a high level, any inference-time sampling approach that generates multiple candidate blocks from $\pi_{\text{target}}$ and uses $r$ to subselect one, has a one-to-one mapping to a different approach which generates candidates from $\pi_{\text{draft}}$ and instead use $S(\cdot|x)$ for subselection. Alternatively, eq. (2) is equivalent to the *score maximization problem*, subject to a KL-penalty *with respect to the generator model*, $\mathbb{E}_{\rho,\pi}[S(Y|\text{tr})] - \frac{1}{\beta}D_{\text{KL}}(\pi\|\pi_{\text{gen}})$.

We will defer a more detailed analysis and interpretation of SUBSAMPLE as an instantiation of sequential monte carlo (SMC) or particle filtering (Rubin, 1987; Naesseth et al., 2019). This method attempts to sample from the distribution in eq. (4) by approximating the partition function, $Z(x) = \mathbb{E}_{Y \sim \pi_{\text{gen}}(\cdot|x)}[\exp(S(Y|x))]$, via computing an empirical estimate over a finite set of "particles".

## 3.2 Theoretical guarantees for SPECS

In this section, we describe theoretical guarantees for SPECS. The nature of result we will establish is that, under certain assumptions, as the beam-width increases, the distribution of output of SPECS converges to the optimal policy of the KL-regularized reward maximization problem (eq. (2)). First we formalize some notation below.

**Notation.** Denote the space of prompts as $\mathcal{X}$, the space of responses as $\mathcal{Y}$. Given a (partial) response $y$, let $(y_1, y_2, \cdots)$ denote its blocks and let $y_{\leq t} = (y_1, \cdots, y_t)$ denote the prefix of the first $t$ blocks. By default, we will let $y_{\leq 0} = \emptyset$. $H$ denotes the horizon: the maximum number of blocks that the model is allowed to generate. We will assume that there exists a target distribution over prompts, denoted $\rho$, and the notation $\mathbb{E}_{\rho, \pi}[\cdot]$ is shorthand for $\mathbb{E}_{X \sim \rho, Y \sim \pi(\cdot|X)}[\cdot]$, where $Y$ is a full response drawn from $\pi$. The goal is to design an algorithm that samples from the policy that maximizes the KL-regularized expected reward (eq. (2)).

Next, we introduce the "idealized" notion of PRM under which we demonstrate guarantees for SPECS. We begin with the notion of an optimal KL-regularized value function (Ziebart et al., 2008; Zhou et al., 2025).

**Definition 1** (Optimal KL-regularized value function). For a policy $\pi$, and any prompt $x \in \mathcal{X}$ and partial trace $y_{\leq t}$, the optimal KL-regularized value of a policy is defined as,

$$V_\beta^\pi(x, y_{\leq t}) = \frac{1}{\beta} \log \left( \mathbb{E}_{Y \sim \pi(\cdot|x, y_{\leq t})} \left[ e^{\beta r(x, Y)} \right] \right)$$

where $Y$ is a completion of the partial trace $y_{\leq t}$. For any full response $y \in \mathcal{Y}$, $V_\beta^\pi(x, y) = r(x, y)$.

**Definition 2** (Idealized Process Reward Model / optimal KL-regularized advantage function). Let $r_{\text{PRM}}$ denote the PRM corresponding to the *KL-regularized advantage function* with respect to some policy $\pi$. Formally, given any prefix of blocks $y_{\leq t-1}$ of length $t-1$ and a subsequent block $y_t$,

$$r_{\text{PRM}}^\pi(y_t|x, y_{\leq t-1}) = V_\beta^\pi(x, y_{\leq t}) - V_\beta^\pi(x, y_{\leq t-1})$$

where $V_\beta^\pi$ is defined in Definition 1. The idealized PRM we consider is $r_{\text{PRM}}^{\text{target}} = r_{\text{PRM}}^\pi$ for $\pi = \pi_{\text{target}}$.

This above notion of PRM has also been used in recent works (Mudgal et al., 2024; Zhou et al., 2025; Brantley et al., 2025) for solving the KL-regularized reward maximization problem. The latter work provides an efficient and scalable approach for learning this PRM from offline datasets, and shows that policies trained by carrying out RL against this PRM perform well empirically, even on low generation budgets. For the purpose of keeping the theoretical presentation cleanest, we analyze SPECS when implemented with an exact version of this PRM. Our main result analyzes SPECS under a subtle modification to make analysis more tractable, which is described in Appendix B.1. This pertains to drawing the number of beams generated per step from a Poisson distribution.

**Theorem 1.** *Assume $n \geq 3$ and suppose SPECS is implemented with the idealized PRM as defined in Definition 2 with beam-width $n$. Then, for reasoning problems over $H$ blocks, in the finite-block length setting, ~~assuming the Poisson sampling model (Assumption 1)~~, the policy $\pi_{SPECS}$ returned by SPECS satisfies,*

$$D_{KL}(\pi_{SPECS}, \pi_\beta^\star) \leq \widetilde{\mathcal{O}}_n \left( H \cdot C_{\text{block}}^2 \frac{e^{2\beta R}}{n^{3/2}} \right) \tag{5}$$

*Here, we assume that the PRM reward range is $[0, R]$. The block-level coverage coefficient $C_{\text{block}}$ is,*

$$C_{\text{block}} = \sup_{t \geq 0} \sup_{y_{\leq t} \in \mathcal{Y}_{\leq t}} \left\| \frac{\pi_{\text{target}}(\cdot|x, y_{\leq t})}{\pi_{\text{draft}}(\cdot|x, y_{\leq t})} \right\|_\infty \left\| \frac{\pi_{\text{draft}}(\cdot|x, y_{\leq t})}{\pi_{\text{target}}(\cdot|x, y_{\leq t})} \right\|_\infty$$

*where $\pi(\cdot|x, y_{\leq t})$ denotes the next-block distribution given the partial trace $y_{\leq t}$ and prompt $x$.*

The proof of this result is involved and deferred to Appendix B.

## 3.3 Robustness of SPECS to misspecified PRMs

Our guarantees for SPECS in the previous section hinge on the assumption that the PRM is idealized, and equal to the optimal KL-regularized advantage function. However, PRMs in practice can often be

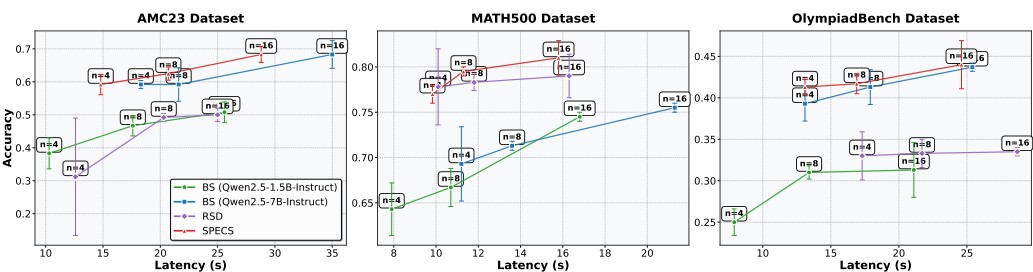

Figure 3: Accuracy vs per-query average latency curves for SPECS and beam search with draft mode, target model and RSD. The error bars show the standard deviation computed over 3 independent runs. The draft model is Qwen2.5-1.5B-Instruct and the target model is Qwen2.5-7B-Instruct.

noisy. In this section, we describe a to be able to benefit from process feedback at test-time, we may weaken this condition by assuming the PRM satisfies some pointwise error guarantee with respect to the optimal KL-regularized advantage function.

**Assumption 1.** Recall the definition of $r_{\text{PRM}}^{\text{target}}$ as defined in Definition 2. Suppose the learner's PRM $r_{\text{PRM}}$ satisfies the conditioned,
$$\|r_{\text{PRM}} - r_{\text{PRM}}^{\text{target}}\|_\infty \leq \varepsilon_{\text{PRM}}.$$
where $\varepsilon_{\text{PRM}} > 0$ is a misspecification parameter.

When SPECS is implemented with this misspecified PRM, we show that the KL divergence of the resulting policy to the optimal policy $\pi_\beta^\star$ degrades gracefully with $\varepsilon_{\text{PRM}}$.

**Theorem 2.** *Assume $n \geq 3$ and suppose SPECS is implemented with a PRM which satisfies Assumption 1, and with beam-width $n$. Then, for reasoning problems over $H$ blocks, in the finite-block length setting, the policy $\pi_{SPECS}$ returned by SPECS satisfies,*
$$D_{KL}(\pi_{SPECS}, \pi_\beta^\star) \leq \widetilde{\mathcal{O}}_n \left( H \cdot C_{\text{block}}^2 \frac{e^{2\beta R}}{n^{3/2}} + \beta H \varepsilon_{PRM} \right)$$

*Here, we assume that the PRM reward range is $[0, R]$.*

Thus, even when the PRM departs from the idealized form we assume earlier in Assumption 1, the algorithm still admits convergence guarantees toward the optimal policy under the *ground-truth reward function*, $\pi_\beta^\star$.

## 4 EXPERIMENTS

In this section, we present empirical evaluation results on the latency-accuracy tradeoff of SPECS and compare it with strong baselines.

### 4.1 EXPERIMENTAL SETUP

**Dataset:** We conduct experiments on three mathematical reasoning benchmarks, AMC23 (Faires & Wells, 2022), MATH500 (Hendrycks et al., 2021) and OlympiadBench (He et al., 2024a). The AMC23 dataset comprises of 40 mathematical reasoning problems, testing concepts in algebra, geometry, number theory, and combinatorics, with an emphasis on precision and strategic problem-solving. MATH500 is a widely used benchmark for evaluating mathematical problem-solving capabilities of language models. OlympiadBench consists of challenging olympiad-level problems, requiring deep reasoning. Following prior works (Qiu et al., 2024; Zhang et al., 2024), we use a randomly selected subset of 100 questions from MATH500 and OlympiadBench datasets to make evaluation faster, and report average performance across multiple seeds.

**Models:** We use several models from the Qwen and Llama families as our base generation models. We evaluate over two draft-target model pairs, Qwen2.5-1.5B-Instruct and Qwen2.5-7B-Instruct from the Qwen family of models (Yang et al., 2024a), as well as Llama3.2-1B-Instruct and Llama-3.1-8B-Instruct (Dubey et al., 2024). These models vary significantly in capability, allowing us to test the scalability of our approach. For reward-guided selection in both beam search and SPECS, we utilize

the Skywork-o1-Open-PRM-Qwen-2.5-7B (He et al., 2024b), a process reward model specifically trained for mathematical reasoning tasks. It is worth pointing out that the generator (draft/target) models have not been specifically finetuned for math reasoning tasks.

**Baselines:** Our primary baseline is standard beam search guided by the same Skywork-o1-Open-PRM-Qwen-2.5-7B process reward model. This involves generating multiple candidate sequences (beams) block-by-block and selecting the highest scoring sequences according to the reward model (Snell et al., 2025; Beeching et al., 2024). We compare SPECS against beam search where the beam generator is the target model (higher latency and accuracy), and is the draft model (lower latency and accuracy).

We also evaluate against reward-guided speculative decoding (RSD) (Liao et al., 2025). RSD is a recent approach that replaces the log-likelihood ratio based rejection sampling in classical speculative decoding with signals from a PRM. A draft model is used to generate a single beam. If the beam is scored low by the PRM, the target model is invoked to regenerate the beam. We implement a multi-beam variant of RSD to provide a much stronger baseline to compete against SPECS: we first decode $n$ beams from the draft model, and pick the beam with highest reward. If the reward of that beam is lower than a threshold, we reject it and sample $n$ beams from the target model instead and pick the highest rewarding beam. We use the rejection threshold explicitly recommended by (Liao et al., 2025) for these experiments, 0.7.

We evaluate performance based on accuracy and latency of the algorithms. Since all of our datasets are in the domain of math reasoning, we calculate the accuracy as the percentage of problems solved correctly. To measure latency, we calculate the wall-clock time required to generate the solution for each problem. We report average latency numbers. Our main focus is the latency-accuracy trade-off, visualizing how accuracy changes as a function of latency acoss different methods and configurations (e.g., varying beam widths).

Some additional analyses including the latency breakdown of SPECS, as well as implementation specifics such as the hardware setup and hyperparameter choices are presented in Appendix D.

### 4.2 RESULTS

We present comparative results on latency and accuracy across the AMC23, MATH500 and Olympiad-Bench datasets in Table 1.

For a fixed beam-width $n$, SPECS is able to achieve comparable or even higher accuracy compared to the strong baseline of the beam search with the large model. We attribute this accuracy gain to the strength that SPECS is able to utilize the ability of both the small and large models. Note that the performance comes with the benefit of up to ∼*18%* latency savings, demonstrating a favorable latency-accuracy trade-off of our proposed SPECS method.

We also report the percentage of steps generated by the large model in the table. The percentage is higher for the more challenging datasets of AMC23, and OlympiadBench compared to MATH500, suggesting that SPECS is able to adapt the usage of large model to the hardness of the questions.

**Pareto-frontier of accuracy vs. latency.** Varying $\tau$, in SPECS (cf. line 9 in Algorithm 1) changes the reward threshold at which the algorithm dynamically switches to generating from the draft model. Higher values of $\tau$ results in a method which switches to the draft model more quickly, resulting in lower latencies and lower accuracies, while lower values of $\tau$ results in blocks being generated for longer by the target model (i.e., higher latency and accuracy). Varying $\tau$ results in an accuracy-latency pareto curve. In Figure 4 we plot this for the MATH500 dataset with the Qwen2.5-1.5B-Instruct draft model and Qwen2.5-7B-Instruct target model. The resulting curve demonstrates a favorable tradeoff for SPECS, the points corresponding to all other baselines (beam search with both target and draft models, and RSD) fall within the convex hull of the Pareto curve traced by SPECS.

Next we present the result of some ablations of SPECS showing the relative benefit of its components: speculative drafting and dynamic switching.

### 4.3 ABLATION STUDIES

**Effect of draft selection score.** First, we compare SPECS with a version of the algorithm where we remove the dynamic switching, and generate all steps from the draft model with soft verification

Table 1: Performance comparison of methods across datasets and beam search settings. Here $n$ denotes per-step beam width, 'Lat. (s)' is average per problem latency in seconds and '% B' represents the percentage of steps generated by the big model B averaged across problems. (S: small model, B: big model). (⋆) we implement a much stronger variant of RSD in our experiments as a competitive baseline to compare SPECS against; more details are provided in Section 4.1.

| Method | $n$ | Acc (%)↑ | Lat (s)↓ | Lat/step (s)↓ | % B | Acc (%)↑ | Lat (s)↓ | Lat/step (s)↓ | % B |
|---|---|---|---|---|---|---|---|---|---|
| | | Qwen2.5 Family | | | | Llama3 Family | | | |
| | | AMC23 | | | | | | | |
| BS(S) | 4 | 38.3 ± 4.7 | 10.3 ± 0.6 | 0.80 ± 0.01 | 0 | 11.7 ± 5.1 | 5.94 ± 0.24 | 0.38 ± 0.15 | 0 |
| BS(B) | 4 | 59.2 ± 1.2 | 18.3 ± 1.5 | 1.46 ± 0.06 | 100 | 26.3 ± 1.3 | 12.7 ± 1.3 | 0.59 ± 0.17 | 100 |
| RSD | 4 | 31.2 ± 17.8 | 12.6 ± 3.8 | 0.82 ± 0.60 | 60.5 ±1.4 | 15.0 ± 7.4 | 8.9 ± 1.1 | 0.66 ±0.79 | 91.8 ±0.6 |
| SPECS | 4 | **59.2 ± 3.1** | 14.8 ± 0.5 | 1.42 ±0.13 | 67.9 ± 1.8 | **26.3 ± 5.3** | 15.4 ± 1.8 | 0.65 ± 0.03 | 96.4 ± 1.2 |
| BS(S) | 8 | 46.7 ± 3.1 | 17.6 ± 2.3 | 1.14 ± 0.01 | 0 | 14.2 ± 4.2 | 11.85 ± 0.22 | 0.61 ± 0.46 | 0 |
| BS(B) | 8 | 59.2 ± 5.1 | 21.6 ± 1.1 | 2.00 ± 0.06 | 100 | 27.5 ±2.0 | 19.1 ±0.96 | 0.84 ±1.39 | 100 |
| RSD | 8 | 49.2 ± 1.2 | 20.3 ± 1.2 | 1.99 ± 4.43 | 61.1 ± 0.3 | 18.3 ± 5.9 | 10.8 ± 1.9 | 0.91 ±0.8 | 90.0 ±1.1 |
| SPECS | 8 | **62.5 ± 2.0** | 20.7 ± 2.0 | 1.94 ±0.11 | 62.8 ± 2.1 | **30.0 ± 2.5** | 28.0 ± 1.2 | 1.04 ± 0.05 | 93.6 ± 1.0 |
| BS(S) | 16 | 50.8 ± 3.1 | 25.6 ± 3.1 | 1.81 ± 0.03 | 0 | 24.2 ± 4.7 | 24.47 ± 1.21 | 1.13 ± 0.95 | 0 |
| BS(B) | 16 | 68.3 ± 4.2 | 35.0 ± 2.6 | 2.80 ± 0.13 | 100 | 32.5 ±4.7 | 32.1 ±2.4 | 1.24 ±1.94 | 100 |
| RSD | 16 | 50.0 ± 2.0 | 25.0 ± 0.8 | 2.56 ± 1.78 | 55.6 ± 2.2 | 20.0 ± 4.1 | 16.3 ± 0.4 | 1.18 ±1.09 | 90.0 ±2.0 |
| SPECS | 16 | **68.3 ± 2.4** | 28.8 ± 3.5 | 2.70 ± 0.05 | 70.3 ± 1.7 | **36.3 ± 3.8** | 44.2 ± 2.9 | 1.56 ± 0.10 | 98.1 ± 1.1 |
| | | MATH500 | | | | | | | |
| BS(S) | 4 | 64.3 ± 2.9 | 7.9 ± 1.0 | 0.65 ± 0.04 | 0 | 23.1 ± 1.5 | 4.82 ± 0.40 | 0.27 ± 3.95 | 0 |
| BS(B) | 4 | 69.3 ± 4.1 | 11.2 ± 3.7 | 1.13 ± 0.01 | 100 | 32.0 ±3.3 | 8.4 ±0.1 | 0.39 ±0.29 | 100 |
| RSD | 4 | **77.8 ± 4.2** | 10.1 ± 0.2 | 1.14 ± 0.52 | 32.3 ±0.6 | **35.7 ± 4.1** | 11.2 ± 0.6 | 0.55 ± 2.15 | 72.7 ±1.1 |
| SPECS | 4 | 77.0 ± 1 | 9.83 ± 0.5 | 1.11 ± 0.14 | 49.0 ± 0.8 | 35.3 ± 3.9 | 10.8 ± 0.8 | 0.40 ± 1.17 | 91.1 ± 0.9 |
| BS(S) | 8 | 66.7 ± 2.1 | 10.7 ± 0.6 | 1.01 ± 0.03 | 0 | 28.3 ± 1.7 | 9.55 ± 0.40 | 0.45 ± 0.68 | 0 |
| BS(B) | 8 | 71.3 ± 0.5 | 13.6 ± 1.9 | 1.51 ± 0.00 | 100 | 33.0 ±2.2 | 12.1 ±0.1 | 0.52 ±0.3 | 100 |
| RSD | 8 | 78.3 ± 0.9 | 11.8 ± 0.3 | 1.37 ± 0.83 | 28.4 ±0.2 | 39.3 ± 2.1 | 15.2 ± 0.5 | 0.71 ± 2.02 | 73.9 ±1.2 |
| SPECS | 8 | **79.5 ± 0.5** | 11.3 ± 0.6 | 1.31 ± 0.56 | 45.5 ± 0.3 | **40.7 ± 2.6** | 15.2 ± 0.74 | 0.57 ± 0.09 | 85.2 ± 1.6 |
| BS(S) | 16 | 74.5 ± 0.5 | 16.8 ± 1.9 | 1.57 ± 0.10 | 0 | 34.3 ± 0.6 | 17.42 ± 0.69 | 0.78 ± 2.42 | 0 |
| BS(B) | 16 | 75.5 ± 0.5 | 21.3 ± 0.5 | 2.15 ± 0.02 | 100 | 39.0 ± 2.2 | 18.9 ± 0.46 | 0.79 ± 0.21 | 100 |
| RSD | 16 | 79.0 ± 2.4 | 16.3 ± 0.4 | 1.84 ± 0.64 | 26.8 ± 0.3 | 40.0 ± 2.2 | 23.0 ± 1.1 | 1.02 ± 1.98 | 70.3 ± 1.3 |
| SPECS | 16 | **81.0 ± 1.9** | 15.8 ± 0.06 | 1.31 ± 0.04 | 45.5 ± 0.8 | **44.0 ± 0.8** | 22.9 ± 6.3 | 0.86 ± 0.19 | 78.1 ± 1.1 |
| | | OlympiadBench | | | | | | | |
| BS(S) | 4 | 25.0 ± 1.6 | 7.9 ± 0.2 | 0.81 ± 0.2 | 0 | 8.0 ± 1.4 | 5.58 ± 0.12 | 0.35 ± 0.26 | 0 |
| BS(B) | 4 | 39.3 ± 2.1 | 13.1 ± 0.3 | 1.32 ± 0.3 | 100 | 17.0 ± 1.2 | 13.0 ± 0.05 | 0.59 ± 0.18 | 100 |
| RSD | 4 | 33.0 ± 2.9 | 17.3 ± 0.4 | 1.60 ± 0.70 | 66.3 ± 0.8 | 7.3 ± 1.7 | 13.1 ± 0.4 | 0.61 ± 0.69 | 95.2 ±0.9 |
| SPECS | 4 | **41.3 ± 0.9** | 13.1 ± 0.3 | 1.30 ± 3.4 | 83.1 ± 0.4 | **18.0 ± 2.8** | 14.9 ± 2.8 | 0.62 ± 0.03 | 98.1 ± 0.2 |
| BS(S) | 8 | 31.0 ± 0.8 | 13.4 ± 0.3 | 1.22 ± 0.3 | 0 | 8.0 ± 3.3 | 11.09 ± 0.49 | 0.59 ± 0.44 | 0 |
| BS(B) | 8 | 41.3 ± 2.1 | 17.9 ± 0.5 | 1.81 ± 0.5 | 100 | 18.3 ± 0.48 | 18.9 ± 0.1 | 0.83 ± 0.14 | 100 |
| RSD | 8 | 33.3 ± 1.7 | 21.7 ± 1.3 | 1.96 ± 1.78 | 65.3 ± 1.2 | 8.0 ± 1.4 | 18.9 ± 0.7 | 0.83 ± 0.56 | 92.9 ±1.1 |
| SPECS | 8 | **41.7 ± 1.2** | 16.9 ± 0.5 | 1.80 ± 1.5 | 80.3 ± 0.4 | **18.5 ± 0.5** | 19.7 ±0.14 | 0.78 ± 0.66 | 94.6 ± 0.1 |
| BS(S) | 16 | 31.3 ± 3.3 | 21.1 ± 1.1 | 2.02 ± 1.1 | 0 | 11.3 ± 1.2 | 23.17 ± 0.73 | 1.06 ± 1.23 | 0 |
| BS(B) | 16 | 43.7 ± 0.5 | 25.4 ± 0.9 | 2.57 ± 0.9 | 100 | 22.0 ± 1.2 | 30.1 ± 0.8 | 1.20 ± 5.3 | 100 |
| RSD | 16 | 33.5 ± 0.5 | 28.7 ± 1.3 | 2.58 ± 13.29 | 61.2 ± 1.3 | 9.5 ± 0.5 | 25.8 ± 0.5 | 1.18 ± 2.71 | 93.2 ± 0.8 |
| SPECS | 16 | **44.0 ± 2.9** | 24.6 ± 1.0 | 2.56 ± 6.2 | 61.5 ± 0.3 | 20.0 ± 5.7 | 31.7 ± 2.4 | 1.21 ± 0.01 | 91.1 ± 1.3 |

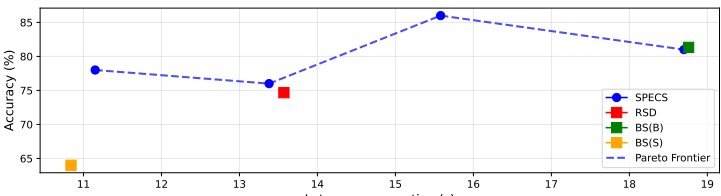

Figure 4: Changing the value of the threshold $\tau$ gives us an accuracy-latency pareto curve for SPECS. As a sanity check, when $\tau \to 1$, the SPECS algorithm collapses to beam search with the large model, and we see that the corresponding points coincide in the figure.

in the sense of eq. (1). This can alternately be seen as instantiating $\pi_{\text{gen}} \leftarrow \pi_{\text{draft}}$, with no dynamic switching. We observe that we obtain a much higher accuracy compared to beam search with the small model while the latency is not much worse (plotted in Table 1). This demonstrates the benefit of our score design in eq. (1). The results of this ablation are shown in Table 2, under the name "no target".

**Effect of dynamic switching.** We also compare with an alternate version where we switch between the draft and target model for generation randomly, in such a way that the average proportion of steps the target model generates matches with the target model usage of SPECS. This results in a method with similar average per-step latencies (although the overall number of steps may vary across methods). We observe that random switching achieves much lower accuracy than dynamic switching based on the reward signal, demonstrating the advantage of our dynamic switching scheme. The results of this ablation are shown in Table 2, under the name "random switch".

Table 2: Performance comparison of ablation studies across different datasets and beam search settings. 'SPECS' is the main method, 'no target' removes target model generation, and 'Random Switch' switches randomly between draft and target model in such a with the same probability as the switching ratio of the target model. Here $n$ denotes the per-step beam width.

| Method | $n$ | Success Rate (%) ↑ | Runtime (s) ↓ | Success Rate (%) ↑ | Runtime (s) ↓ | Success Rate (%) ↑ | Runtime (s) ↓ |
|---|---|---|---|---|---|---|---|
| | | AMC23 | | MATH500 | | OlympiadBench | |
| SPECS | 4 | 59.2 | 14.80 | 77.0 | 9.83 | 41.3 | 13.10 |
| No Target | 4 | 51.2 | 13.68 | 65.0 | 7.45 | 41.1 | 13.00 |
| Random Switch | 4 | 42.2 | 10.86 | 70.0 | 8.59 | 30.1 | 9.43 |
| SPECS | 8 | 62.5 | 20.70 | 79.5 | 11.3 | 41.7 | 16.90 |
| No Target | 8 | 56.2 | 19.69 | 66.0 | 8.15 | 39.0 | 16.24 |
| Random Switch | 8 | 50.6 | 15.45 | 70.0 | 12.13 | 33.5 | 13.86 |
| SPECS | 16 | 68.3 | 28.80 | 81.0 | 11.748 | 44.0 | 24.60 |
| No Target | 16 | 63.8 | 26.72 | 73.0 | 11.6 | 39.5 | 23.84 |
| Random Switch | 16 | 54.7 | 23.60 | 74.0 | 11.3 | 38.4 | 21.83 |

## 5 CONCLUSION

In this work, we address the critical challenge of balancing latency and accuracy in test-time scaling for large language models. We propose SPECS, a latency-aware test-time scaling method for large language models (LLMs) that significantly improves the latency-accuracy tradeoff through speculative drafting with a fast draft model. We propose novel integration strategies to merge the draft model, target model, and the PRM, including reward-guided soft verification and a reward-based switching mechanism, to dynamically balance computational resources with response latency. Empirically, we evaluate SPECS across the MATH500, AMC23, and OlympiadBench datasets, demonstrating its superior performance. SPECS consistently achieves accuracy comparable to or exceeding traditional beam search methods while delivering substantial reductions in latency - up to approximately 18%. Additionally, our theoretical analysis confirms that SPECS converges to the optimal solution of a KL-regularized reinforcement learning objective as the beam width increases.

## 6 REPRODUCIBILITY STATEMENT

To ensure reproducibility, we provide detailed descriptions of our algorithms and include pseudocode for all key components in the main paper. Our codebase, containing the essential implementation details, is included as supplementary material, and we plan to release the full polished code along with complete experiment logs upon acceptance. We also share the details of our hyperparameters search and selected hyperparameters in the Appendix. All experiments were conducted on publicly available datasets, which we describe in the paper, and repeated across multiple random seeds to confirm the stability and reproducibility of our results.

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

## A   RELATED WORK

**Reward-guided search.** Our algorithm is related to the general literature of process reward modeling (Uesato et al., 2022; Lightman et al., 2023; Wang et al., 2024; Snell et al., 2025; Beeching et al., 2024; Zheng et al., 2024; Zhang et al., 2025b) and reward-guided search (Mudgal et al., 2024; Beeching et al., 2024; Sun et al., 2024; Qiu et al., 2024; Snell et al., 2025). Among these, the closest to our is reward-guided speculative decoding (Liao et al., 2025), which also uses a draft model to propose each reasoning step and uses a process reward model to decide on whether to keep the proposal. SPECS generalizes this by proposing multiple candidate reasoning and combine signals from both the large model and reward model to decide on the candidate reasoning step to keep. Empirically, we observe our method achieves better latency-accuracy tradeoff compared to reward-guided speculative decoding.

Concurrent work of Geuter et al. (2025) also considers soft draft verification by sampling from an exponentially tilted distribution. While their algorithm closes part of the accuracy gap between the draft model and the target model, our algorithm is able to achieve even higher accuracy than the target model alone. Moreover, we also provide convergence guarantees on the multi-step version of our algorithm.

We would like to mention that existing work that utilizes speculative drafts to reduce latency all start with generations from a speculative model, and switches to a larger when the reward signal is low. While our algorithm starts with the large model and switches to speculative drafts only when the reward signal is high. This approach reduces the time wasted for speculative drafting for low-reward traces, resulting in overall better latency-accuracy tradeoff. This builds on the novel insight that the accuracy gap between a small model and a large model is small for these traces, which could be of independent interest.

**Tree-based speculative decoding.** Obtaining one sequence from multiple draft sequences has been studied in the speculative decoding literature with the goal of maximizing the likelihood/length of draft tokens accepted while maintaining the same distribution as the large model (Sun et al., 2023; Miao et al., 2024; Cai et al., 2024; Wang et al., 2025a; Chen et al., 2024). Compared to these, the goal of our soft verification algorithm is to maximize the success rate of downstream applications such as math-reasoning as guided by both the target model and the reward model. Other soft verification methods for speculative decoding (Wang et al., 2025b) consider token-level verification while we focus on step-level candidate selection.

**Formal analysis of best-of-$N$ variants.** The theoretical analysis of our soft verification algorithm is related to (Verdun et al., 2025; Huang et al., 2025), which consider alternative candidate selection methods other than Top-1 selection. Our analysis captures the convergence rate in presence of draft models and multiple decoding steps.

**Cascading.** Cascading techniques (Dohan et al., 2022; Jitkrittum et al., 2023; Yue et al., 2024; Narasimhan et al., 2024) also utilize multiple language models and decide on the model to used based on certain deferral rule. The switching step in our algorithm can be viewed as imposing a deferral rule based on small, target, and reward models. Additionally, our deferral rule operates at the step-level while previous work usually focuses on individual tokens or the entire generation.

## B   THEORY

In this section, we describe theoretical guarantees for SPECS. The nature of result we will establish is that, under certain assumptions, as the beam-width increases, the distribution of output of SPECS converges to the optimal policy of the reward maximization problem with KL regularization toward the target model. We first introduce some relevant notation.

**Notation.**   Denote the space of prompts as $\mathcal{X}$, the space of responses as $\mathcal{Y}$, the draft model $\pi_{\text{draft}} : \mathcal{X} \to \Delta_{\mathcal{Y}}$ and the target model $\pi_{\text{target}} : \mathcal{X} \to \Delta_{\mathcal{Y}}$. Prompts and responses are composed of sequences of tokens, and responses are assumed to be complete: ending in a special "End-Of-Sentence" (EOS) token. We will assume that there exists a target distribution over prompts, denoted $\rho$, and the notation $\mathbb{E}_{\rho,\pi}[\cdot]$ is shorthand for $\mathbb{E}_{X \sim \rho, Y \sim \pi(\cdot|X)}[\cdot]$, where the random variable $Y$ corresponds to full responses drawn from $\pi$ under the prompt $X$. The objective is to find a policy which maximizes the expected

reward, while staying close to the target model (as induced by a KL regularization penalty toward $\pi_{\text{target}}$),

$$\mathcal{L}_\beta(\pi) = \mathbb{E}_{\rho,\pi}[r(y|x)] - \frac{1}{\beta}\mathbb{E}_\rho[D_{\text{KL}}(\pi(\cdot|x)\|\pi_{\text{target}}(\cdot|x))] \tag{6}$$

For two models $\pi_1$ and $\pi_2$, we use the notation $D_{\text{KL}}(\pi_1\|\pi_2) = \mathbb{E}_\rho[D_{\text{KL}}(\pi_1(\cdot|x)\|\pi_2(\cdot|x))]$.

Prior to analyzing the algorithm, we begin with a well-known connection between the KL regularized reward maximization objective and the KL divergence between the learnt policy and the optimal aligned policy Korbak et al. (2022b;a); Yang et al. (2024b). We include the proof for the sake of completeness.

**Lemma 1** (Equivalence of KL minimization and KL-regularized reward maximization). *For any policy $\pi$,*

$$\mathcal{L}_\beta(\pi_\beta^\star) - \mathcal{L}_\beta(\pi) = D_{KL}(\pi\|\pi_\beta^\star) \tag{7}$$

*where $\pi_\beta^\star$ is the maximizer of $\mathcal{L}_\beta(\pi)$.*

*Proof.* By Yang et al. (2024b), the optimal solution to eq. (2), $\pi_\beta^\star$ takes the structural form,

$$\pi_\beta^\star(y|x) = \frac{\pi_{\text{target}}(y|x)e^{\beta r(y|x)}}{Z(x)}$$

where $Z(x) = \mathbb{E}_{y\sim\pi_{\text{target}}(y|x)}[e^{\beta r(y|x)}]$. Thus,

$$\mathcal{L}_\beta(\pi) = \mathbb{E}_{\rho,\pi}[r(y|x)] - \frac{1}{\beta}\mathbb{E}_\rho[\mathbb{E}_{y\sim\pi(\cdot|x)}[\log(\pi(y|x)/\pi_{\text{target}}(y|x))]]$$

$$= -\frac{1}{\beta}\mathbb{E}_\rho\big[\mathbb{E}_{y\sim\pi(\cdot|x)}\big[\log\big(\pi(y|x)/(\pi_{\text{target}}(y|x)e^{\beta(r(y|x))})\big)\big]\big]$$

$$= -\frac{1}{\beta}\mathbb{E}_\rho[\log(Z(x))] - \frac{1}{\beta}\mathbb{E}_\rho\big[\mathbb{E}_{y\sim\pi(\cdot|x)}\big[\log\big(\pi(y|x)/\pi_\beta^\star(y|x)\big)\big]\big]$$

$$= \mathcal{L}_\beta(\pi_\beta^\star) - \frac{1}{\beta}\mathbb{E}_\rho\big[D_{\text{KL}}(\pi(\cdot|x)\|\pi_\beta^\star(\cdot|x))\big]$$

Rearranging completes the proof. $\qquad\square$

Therefore, up to an additive error term that does not depend on $\pi$, the loss $\mathcal{L}_\beta(\pi)$ measures the KL divergence between the distribution over responses induced by $\pi$ and $\pi_\beta^\star$ under prompts generated from $\rho$.

The purpose of introducing this result is to motivate the reason for choosing the error measure we study throughout the paper as the *reverse* KL divergence, rather than the forward KL divergence. The former is closely connected with the KL regularized reward maximization objective (it measures the suboptimality gap), while the latter is not so clearly connected. Indeed, these objectives penalize the learnt policy differently: the latter more strongly penalizes policies for placing too high mass on certain responses while the former more strongly penalizes policies which place too low mass on responses.

### B.1 WARM-UP: ANALYSIS OF SPECS IN THE INFINITE-BLOCK LENGTH REGIME

In this section we will build up our theoretical understanding of SPECS by considering the infinite block-length setting. In this setting, the drafts generated by the draft model are assumed to be full responses. Here, the algorithm simply selects one candidate among a number of responses using the SUBSAMPLE subroutine.

**Definition 3** (Infinite block-length regime). The infinite block-length regime corresponds to taking $\gamma \to \infty$. Operationally, the draft/target model generates full responses and the process reward model collapses to an outcome reward model.

In this section we will establish a convergence rate of $O(1/n^2)$ for SPECS in the infinite block-length setting. Increasing $n$ scales the amount of (parallel) test-time compute used by the algorithm.

**Theorem 3** (Guarantee for SPECS in the infinite-block length regime). *Suppose* $n \geq 3$ *and assume that the reward function lies in the range* $[0, R]$ *pointwise.* *Then, ~~assuming the Poisson sampling model (Assumption 1)~~, the policy* $\pi_{SPECS}$ *returned by SPECS in the infinite block-length setting (Definition 3), , satisfies,*

$$D_{KL}(\pi_{SPECS} \| \pi_\beta^\star) \leq \widetilde{\mathcal{O}}_n \left( C_{\text{seq}}^2 \frac{e^{2\beta R}}{n^{3/2}} \right) \tag{8}$$

*where, the "sequence-level coverage coefficient"* $C_{\text{seq}}$ *is,*

$$C_{\text{seq}} = \left\| \frac{\pi_{\text{target}}(\cdot|x)}{\pi_{\text{draft}}(\cdot|x)} \right\|_\infty \left\| \frac{\pi_{\text{draft}}(\cdot|x)}{\pi_{\text{target}}(\cdot|x)} \right\|_\infty. \tag{9}$$

*Remark* 1. In our analysis we establish a stronger result, where the LHS of Eq. (8) is replaced by $\log\left( 1 + D_{\chi^2}(\pi_{\text{SPECS}} \| \pi_\beta^\star) \right)$.

The proof of this result follows as a consequence of Corollary 2 which we will introduce later. In the infinite block-length regime, SPECS runs Algorithm 2 once with the generator model $\pi_{\text{gen}} \leftarrow \pi_{\text{target}}$. In the next section, we will prove a guarantee on the distribution over responses / steps induced SUBSAMPLE routine we consider in Algorithm 2 as an instantiation of sequential monte carlo (SMC) Rubin (1987); Naesseth et al. (2019), also known by the name particle filtering in the literature Gordon et al. (1993). In the next section, we will introduce SMC and prove a distributional closeness result for the resulting policy. In turn, this will imply the proof of Theorem 3 and provide a starting point for proving Theorem 1.

### B.2 SEQUENTIAL MONTE CARLO: SAMPLING BY APPROXIMATING THE PARTITION FUNCTION

Given $N$ responses drawn from the generator model $\pi_{\text{gen}}(\cdot|x)$, the partition function $Z(x)$ in eq. (4) can be approximated by a discrete summation,

$$Z(x) = \mathbb{E}_{Y \sim \pi_{\text{gen}}(\cdot|x)}[e^{S(Y|x)}] \approx \frac{1}{N} \sum_{i=1}^{N} e^{S(Y_i|x)} = \widehat{Z}(x).$$

SMC uses this approximation to sample from the target distribution, $\pi_{\text{gen}}(\cdot|x)e^{S(\cdot|x)}/Z(x)$.

**Definition 4** (Sequential Monte Carlo (SMC)). Given a prompt $x$, SMC generates a set of draft responses $\mathbf{Y}_{\text{draft}} = \{Y_i\}_{i=1}^{N}$ i.i.d. from $\pi_{\text{gen}}(\cdot|x)$. Then an index $i$ is sampled with probability proportional to $e^{S(Y_i|x)}$ and $Y_i$ is returned. Denoting $Y^\star$ as the output of the procedure,

$$\Pr(Y^\star = Y_i) = \frac{e^{S(Y_i|x)}}{Z_{\mathbf{Y}_{\text{draft}}}(x)},$$

$$\text{where, } Z_{\mathbf{Y}_{\text{draft}}}(x) = \sum_{j \in [n]} e^{S(Y_j|x)}.$$

The policy, $\pi_{\text{SMC},N}$, is the induced distribution over responses, averaged over the randomness of sampling $\mathbf{Y}_{\text{draft}}$.

Let us first introduce some notation to make the presentation simpler.

**Definition 5** (Exponentiated score). For the sake of brevity, define the exponentiated score,

$$\phi_\beta(y|x) = \frac{\pi_{\text{target}}(y|x)}{\pi_{\text{gen}}(y|x)} e^{\beta r(y|x)}.$$

Furthermore, define $\phi_{\max} = \max_{x \in \mathcal{X}, y \in \mathcal{Y}} \phi_\beta(y|x)$ and $\phi_{\min} = \min_{x \in \mathcal{X}, y \in \mathcal{Y}} \phi_\beta(y|x)$. Note that $\phi_\beta$ is simply the exponentiated version of the score $S(y|x)$ defined in eq. (1).

To aid our analysis, we will establish guarantees for SPECS which bounds the KL divergence of with the optimal policy, when the number of drafts generated (i.e., the beam-width) is randomized and distributed according to a Poisson random variable with mean $n$. We will establish upper bounds on $\mathbb{E}[D_{\text{KL}}(\pi_{\text{SMC},N} \| \pi_\beta^\star)]$ where $N \sim \text{Poi}(n)|_{>0}$. Note that since $\Pr(N = n) \gtrsim 1/n$ for $N \sim \text{Poi}(n)|_{>0}$ so, these results can be translated into an upper bound on $D_{\text{KL}}(\pi_{\text{SMC},n} \| \pi_\beta^\star)$ for any fixed value of $n$, by multiplying by an additional $\sqrt{n}$ factor.

**Theorem 4.** *Fix any $n \geq 3$. In the infinite block-length setting, the policy $\pi_{SMC,N}$ induced by SMC (Definition 4). satisfies,*

$$\mathbb{E}_{N \sim \text{Poi}(n)|_{>0}}[D_{KL}(\pi_{SMC,N} \| \pi_\beta^\star)] \leq \mathcal{O}\left(\frac{\log^2(n)}{n^2}\left(\frac{\phi_{\max}}{\phi_{\min}}\right)^2\right).$$

*$\phi_{\max}$ and $\phi_{\min}$ are defined in Definition 5.*

This result presents a guarantee for SMC where the number of beams generated is drawn from a Poisson distribution. Poisson random variables are tightly concentrated around their mean and the above result implies as a corollary that for any $n \geq 3$,

$$\min_{N \leq n} D_{\text{KL}}(\pi_{\text{SMC},N} \| \pi_\beta^\star) \leq \mathcal{O}\left(\frac{\log^2(n)}{n^2}\left(\frac{\phi_{\max}}{\phi_{\min}}\right)^2\right).$$

Similarly, as mentioned previously, note that for $N \sim \text{Poi}(n)|_{>0}$, $\Pr(N = n) \gtrsim 1/\sqrt{n}$. Therefore, by an application of Markov's inequality, we also arrive at the following bound for any fixed value of $n \geq 3$,

$$D_{\text{KL}}(\pi_{\text{SMC},n} \| \pi_\beta^\star) \leq \mathcal{O}\left(\frac{\log^2(n)}{n^{3/2}}\left(\frac{\phi_{\max}}{\phi_{\min}}\right)^2\right). \tag{10}$$

This establishes an upper bound for SMC which applies for any fixed beam width.

Since it is more involved, we defer the proof of Theorem 4 to Appendix C. We have the following two corollaries of Theorem 4 by instantiating $\pi_{\text{gen}}$ differently.

**Corollary 1** (Target sampling). *Suppose $n \geq 3$ and that drafts are generated from $\pi_{\text{target}}$ (i.e., we choose $\pi_{\text{gen}} \leftarrow \pi_{\text{target}}$). Assuming rewards lie in the range $[0, R]$, in the infinite block-length setting (Definition 3),*

$$\mathbb{E}_{N \sim \text{Poi}(n)|_{>0}}[D_{KL}(\pi_{SMC,N} \| \pi_\beta^\star)] \leq \widetilde{\mathcal{O}}_n\left(\frac{e^{2\beta R}}{n^2}\right)$$

*By an application of Markov's inequality, we have that for any fixed $n \geq 3$,*

$$D_{KL}(\pi_{SMC,n} \| \pi_\beta^\star) \leq \widetilde{\mathcal{O}}_n\left(\frac{e^{2\beta R}}{n^2}\right)$$

Note that SPECS in the infinite-block length regime is precisely the same as $\pi_{\text{SMC},N}$ where the generator $\pi_{\text{gen}} \leftarrow \pi_{\text{target}}$. This implies the statement of Theorem 3.

**Corollary 2** (Speculative sampling). *Suppose $n \geq 3$ and that drafts are generated from $\pi_{\text{draft}}$ (i.e., we choose $\pi_{\text{gen}} \leftarrow \pi_{\text{draft}}$). Then, in the infinite block-length setting (Definition 3), assuming rewards lie in the range $[0, R]$,*

$$\mathbb{E}_{N \sim \text{Poi}(n)|_{>0}}[D_{KL}(\pi_{SMC,N} \| \pi_\beta^\star)] \leq \widetilde{\mathcal{O}}_n\left(C_{\text{seq}}^2 \frac{e^{2\beta R}}{n^2}\right)$$

*where $C_{\text{seq}} = \|\pi_{\text{target}}(\cdot|x)/\pi_{\text{draft}}(\cdot|x)\|_\infty \|\pi_{\text{draft}}(\cdot|x)/\pi_{\text{target}}(\cdot|x)\|_\infty$, as defined in Equation (9). By an application of Markov's inequality, we have that for any fixed $n \geq 3$,*

$$D_{KL}(\pi_{SMC,n} \| \pi_\beta^\star) \leq \widetilde{\mathcal{O}}_n\left(C_{\text{seq}}^2 \frac{e^{2\beta R}}{n^{3/2}}\right)$$

*Remark* 2. In all three prior results, Theorem 4 and corollaries 1 and 2, the proof will argue the stronger upper bound which replaces the KL divergence in the LHS by the strictly larger quantity, $\log\left(1 + D_{\chi^2}(\pi_{\text{SMC},N} \| \pi_\beta^\star)\right)$.

When the target and draft models are identical, then $C_{\text{seq}} = 1$. We also show that the rate of decay in Corollary 2 is essentially tight when the target and draft models are identical.

**Theorem 5.** *There exists a target model $\pi_{\text{target}}$ and reward model over a 2-ary response space $\mathcal{Y} = \{y_0, y_1\}$, with rewards lying in the range $[0, R]$, such that, the policy $\pi_{SMC,N}$ returned by sampling by SMC (Definition 4) in the infinite block-length setting (Definition 3) with $\pi_{\text{gen}} \leftarrow \pi_{\text{target}}$,*

$$\mathbb{E}_{N \sim Poi(n)|_{>0}}[D_{KL}(\pi_{SMC,N} \| \pi_\beta^\star)] \geq \Omega_n \left( \frac{e^{2\beta R}}{n^2} \right),$$

*assuming that $n \geq e^{\beta R}$.*

### B.3 FINITE-BLOCK LENGTH: ANALYSIS OF SPECS WITH IDEALIZED PROCESS FEEDBACK

In this section we extend the previous discussion to the finite-block setting. In this case, SPECS (Algorithm 1) is iterative and the reward model considered in SUBSAMPLE (Algorithm 2) must be implemented with a process reward model (PRM) which is trained to provide block-level feedback. Furthemore, control of draft generation is shared between the draft and target models additionally through dynamic switching.

In the infinite block-length setting, the guarantee of SPECS depended on a strong notion of coverage, $C_{\texttt{seq}} = \|\pi_{\text{target}}(\cdot|x)/\pi_{\text{draft}}(\cdot|x)\|_\infty \|\pi_{\text{draft}}(\cdot|x)/\pi_{\text{target}}(\cdot|x)\|_\infty$, where the distributions are over full responses, a quantity which may scale exponentially with $H$. In the finite block-length setting, our main result, Theorem 1 shows that with access to a PRM in the sense of Definition 2 allows guarantees that scale polynomially with the horizon $H$, where the dependency on $C_{\texttt{seq}}$ is relaxed to the maximum "per-block" ratio of densities between the target and draft models.

We will reiterate the definitions of the process reward model we consider. In practice this model is trained and inexact, but for the purpose of a clean theoretical discussion we avoid this challenge and discuss the case where the process reward is exact. The following two definitions instantiate the PRM.

**Definition 6** (Optimal KL-regularized value function: restatement of Definition 1). For a policy $\pi$, and any prompt $x \in \mathcal{X}$ and partial trace $y_{\leq t}$, the optimal KL-regularized value of a policy is defined as,

$$V_\beta^\pi(x, y_{\leq t}) = \frac{1}{\beta} \log \left( \mathbb{E}_{Y \sim \pi(\cdot|x, y_{\leq t})} \left[ e^{\beta r(x, Y)} \right] \right)$$

where $Y$ is a full completion of the partial response $y_{\leq t}$. Note that for any full response $y \in \mathcal{Y}$, $V_\beta^\pi(x, y) = r(x, y)$.

**Definition 7** (Idealized Process Reward Model: restatement of Definition 2). Let $r_{\mathsf{PRM}}^\pi$ denote the PRM corresponding to the optimal KL-regularized advantage function with respect to some policy $\pi$. Formally, given any prefix of tokens / blocks $y_{\leq t-1}$ of length $t-1$ and a subsequent token / block $y_t$,

$$r_{\mathsf{PRM}}^\pi(y_t|x, y_{\leq t-1}) = V_\beta^\pi(x, y_{\leq t}) - V_\beta^\pi(x, y_{\leq t-1})$$

where, the KL-regularized value function $V_\beta^\pi$ is defined in Definition 1. The cumulative advantage till block $t$ is denoted, $r_{\mathsf{PRM}}^\pi(y_{\leq t}|x) = \sum_{t'=1}^t r_{\mathsf{PRM}}^\pi(y_{t'}|x, y_{\leq t'-1})$. The idealized PRM we consider is $r_{\mathsf{PRM}}^{\text{target}} = r_{\mathsf{PRM}}^\pi$ for $\pi = \pi_{\text{target}}$.

Our first result will be to establish that the optimal aligned model can be decomposed into a sequence of models corresponding to solving "one-block" regularized reward maximization problems over the reward function induced by $r_{\mathsf{PRM}}^{\text{target}}$.

**Lemma 2.** *The optimal aligned model over full responses can be written as a product of conditional per-block models,*

$$\pi_\beta^\star(y_1, \cdots, y_t|x) = \prod_{t=1}^H \pi_\beta^\star(y_t|x, y_{\leq t-1}), \tag{11}$$

*where, we define,*

$$\pi_\beta^\star(y_t|x, y_{\leq t-1}) \propto \pi_{\text{target}}(y_t|x, y_{\leq t-1}) e^{\beta[r_{\mathsf{PRM}}^{target}(y_t|x, y_{\leq t-1})]}$$

*Is the solution to a one-block KL-regularized reward maximization problem. Namely, for every prompt $x$ and partial trace $y_{\leq t-1}$, $\pi_\beta^\star(\cdot|x, y_{\leq t-1})$ is supported over one-block completions and is the solution to the problem,*

$$\min\left\{\mathbb{E}_{Y_t \sim \pi(\cdot|x,y_{\leq t-1})}\left[r_{\mathrm{PRM}}^{target}(y_t|x, y_{t-1})\right] - \frac{1}{\beta}D_{KL}(\pi(\cdot|x, y_{\leq t-1}), \pi_{\mathrm{target}}(\cdot|x, y_{\leq t-1}))\right\}$$

*where $\pi : \mathcal{X} \times \mathcal{Y}_{\leq t-1} \to \Delta_{\mathcal{Y}_t}$.*

*Proof.* By definition, the optimal aligned policy can be written down as,

$$\pi_\beta^\star(y|x) = \frac{\pi_{\mathrm{target}}(y|x)e^{\beta r(y|x)}}{\mathbb{E}_{Y \sim \pi_{\mathrm{target}}(\cdot|x)}\left[e^{\beta r(Y|x)}\right]}$$

$$= \pi_{\mathrm{target}}(y|x)e^{\beta[V_\beta^{\pi_{\mathrm{target}}}(x,y) - V_\beta^{\pi_{\mathrm{target}}}(x,\emptyset)]}$$

$$= \pi_{\mathrm{target}}(y|x)\prod_{t'=1}^{H} e^{\beta[V_\beta^{\pi_{\mathrm{target}}}(x,y_{\leq t'}) - V_\beta^{\pi_{\mathrm{target}}}(x,y_{\leq t'-1})]}$$

$$= \prod_{t=1}^{H} \frac{\pi_{\mathrm{target}}(y_t|x, y_{\leq t-1})e^{\beta[V_\beta^{\pi_{\mathrm{target}}}(x,y_{\leq t})]}}{\mathbb{E}_{Y_t \sim \pi_{\mathrm{target}}(\cdot|x,y_{\leq t-1})}\left[e^{\beta V_\beta^{\pi_{\mathrm{target}}}(x,[y_{\leq t-1},Y_t])}\right]}$$

$$= \prod_{t=1}^{H} \frac{\pi_{\mathrm{target}}(y_t|x, y_{\leq t-1})e^{\beta[r_{\mathrm{PRM}}^{\mathrm{target}}(y_t|x,y_{\leq t-1})]}}{\mathbb{E}_{Y_t \sim \pi_{\mathrm{target}}(\cdot|x,y_{\leq t-1})}\left[e^{\beta r_{\mathrm{PRM}}^{\mathrm{target}}(Y_t|x,y_{\leq t-1})}\right]}.$$

$\square$

Next, we show that policies which can guarantee that they realize a one-block distribution which is close to $\pi_\beta^\star(\cdot|x, y_{\leq t-1})$ for every $t$ correspond to policies which is close to $\pi_\beta^\star$ over full responses in the aggregate.

**Lemma 3** (Local-to-global guarantee). *Consider a policy with next-block distribution given by $\pi(\cdot|x, y_{t-1})$ which satisfies the following guarantee: for every prompt $x$ and prefix $y_{t-1}$,*

$$D_{\chi^2}(\pi(\cdot|x, y_{\leq t-1}), \pi_\beta^\star(\cdot|x, y_{\leq t-1})) \leq \varepsilon.$$

*where the induced distributions $\pi(\cdot|x, y_{\leq t-1})$ and $\pi_\beta^\star(\cdot|x, y_{\leq t-1})$ are one-block completions of the partial response $(x, y_{\leq t-1})$. Then, we have that,*

$$1 + D_{\chi^2}(\pi, \pi_\beta^\star) \leq (1+\varepsilon)^H$$

*Proof.* By definition

$$1 + D_{\chi^2}(\pi\|\pi_\beta^\star)$$

$$= \mathbb{E}_\rho\left[\mathbb{E}_{Y \sim \pi_\beta^\star(\cdot|x)}\left[\left(\frac{\pi(Y|x)}{\pi_\beta^\star(Y|x)}\right)^2\right]\right]$$

$$= \mathbb{E}_\rho\left[\mathbb{E}_{Y \sim \pi_\beta^\star(\cdot|x)}\left[\prod_{t=1}^{H}\left(\frac{\pi(Y_t|x, Y_{\leq t-1})}{\pi_\beta^\star(Y_t|x, Y_{\leq t-1})}\right)^2\right]\right]$$

$$= \mathbb{E}_\rho\left[\mathbb{E}_{Y_{\leq H-1} \sim \pi_\beta^\star(\cdot|x)}\left[[1 + D_{\chi^2}(\pi(\cdot|x, Y_{\leq H-1})\|\pi_\beta^\star(\cdot|x, Y_{\leq H-1}))] \cdot \prod_{t=1}^{H-1}\left(\frac{\pi(Y_t|x, Y_{\leq t-1})}{\pi_\beta^\star(Y_t|x, Y_{\leq t-1})}\right)^2\right]\right]$$

$$\overset{(a)}{\leq} (1+\varepsilon)\mathbb{E}_\rho\left[\mathbb{E}_{Y_{\leq H-1} \sim \pi_\beta^\star(\cdot|x)}\left[\prod_{t=1}^{H-1}\left(\frac{\pi(Y_t|x, Y_{\leq t-1})}{\pi_\beta^\star(Y_t|x, Y_{\leq t-1})}\right)^2\right]\right],$$

where $(a)$ follows from the local assumption on $\pi$. By recursing on the RHS, we arrive at the statement of the lemma. $\square$

Combining this recursion with our guarantees for SPECS in the infinite block-length setting, we prove the main result Theorem 1 next.

**Proof of Theorem 1.** Consider some partial trace $y_{\leq t-1}$ and let us consider the process of generating the $t^{\text{th}}$ block. Let $Z_t \in \{0, 1\}$ denote the random variable which is a measurable function of $y_{\leq t-1}$ which determines whether $\pi_{\text{draft}}$ generates the drafts or $\pi_{\text{target}}$ generates drafts in iteration $t$.

By the observation in Remark 2 for the infinite block-length instantiation of the SPECS policy $\pi_{\text{SPECS}}$, the following local guarantee is also true when instantiated with $r = r_{\text{PRM}}^{\text{target}}$ and used to generate a single block: given any prompt $x$ and partial trace $y_{\leq t-1}$,

$$D_{\chi^2}(\pi_{\text{SMC},n}(\cdot|x, y_{\leq t-1})|_{\{Z_t=0\}} \| \pi_{\beta}^{\star}(\cdot|x, y_{\leq t-1})) \leq \exp\left(\widetilde{\mathcal{O}}_n\left(\frac{e^{2\beta R}}{n^{3/2}}\right)\right) - 1$$

$$D_{\chi^2}(\pi_{\text{SMC},n}(\cdot|x, y_{\leq t-1})|_{\{Z_t=1\}} \| \pi_{\beta}^{\star}(\cdot|x, y_{\leq t-1})) \leq \exp\left(\widetilde{\mathcal{O}}_n\left(C_{\text{block}}^2 \frac{e^{2\beta R}}{n^{3/2}}\right)\right) - 1$$

Note in particular that the second guarantee only depends on $C_{\text{block}}$ by virtue of the fact that we only generate a single block and not entire responses ($\pi_{\text{SPECS}}(\cdot|x, y_{t-1})$ is a distribution over single blocks). By the convexity of the $\chi^2$-divergence, we have that $\pi_{\text{SPECS}}(\cdot|x, y_{\leq t-1})$, which is a mixture between $\pi_{\text{SMC},n}(\cdot|x, y_{\leq t-1})$ with $\pi_{\text{gen}} \leftarrow \pi_{\text{draft}}$ (i.e., $Z_t = 0$) and $\pi_{\text{SMC},n}(\cdot|x, y_{\leq t-1})$ with $\pi_{\text{gen}} \leftarrow \pi_{\text{target}}$ (i.e., $Z_t = 1$) satisfies,

$$D_{\chi^2}(\pi_{\text{SPECS}}(\cdot|x, y_{\leq t-1}) \| \pi_{\beta}^{\star}(\cdot|x, y_{\leq t-1})) \leq \exp\left(\widetilde{\mathcal{O}}_n\left(C_{\text{block}}^2 \frac{e^{2\beta R}}{n^{3/2}}\right)\right) - 1$$

The proof finishes by combining these guarantees across $t$ via Lemma 3 to give the inequality,

$$1 + D_{\chi^2}(\pi_{\text{SPECS}} \| \pi_{\beta}^{\star}) \leq \exp\left(\widetilde{\mathcal{O}}_n\left(H \cdot C_{\text{block}}^2 \frac{e^{2\beta R}}{n^{3/2}}\right)\right)$$

Using the inequality, $D_{\text{KL}}(\pi_{\text{SPECS}} \| \pi_{\beta}^{\star}) \leq \log\left(1 + D_{\chi^2}(\pi_{\text{SPECS}} \| \pi_{\beta}^{\star})\right)$ completes the proof.

**Proof of Theorem 2.** With an arbitrary choice of PRM $r_{\text{PRM}}$, it is a simple modification of the proof of Theorem 4 to see that,

$$D_{\text{KL}}(\pi_{\text{SPECS}} \| \pi_{\beta,\text{PRM}}^{\star}) \leq \widetilde{\mathcal{O}}_n\left(H \cdot C_{\text{block}}^2 \frac{e^{2\beta R}}{n^{3/2}}\right)$$

Where, similar to eq. (11),

$$\pi_{\beta,\text{PRM}}^{\star}(y|x) \triangleq \prod_{t=1}^{H} \pi_{\beta,\text{PRM}}^{\star}(y_t|x, y_{\leq t-1})$$

Where, we define,

$$\pi_{\beta,\text{PRM}}^{\star}(y_t|x, y_{\leq t-1}) \propto \pi_{\text{target}}(y_t|x, y_{\leq t-1}) e^{\beta[r_{\text{PRM}}(y_t|x, y_{\leq t-1})]}$$

The statement of theorem 2 will follow from the following inequality,

$$D_{\text{KL}}(\pi_{\text{SPECS}} \| \pi_{\beta}^{\star}) \leq D_{\text{KL}}(\pi_{\text{SPECS}} \| \pi_{\beta,\text{PRM}}^{\star}) + \log\left(\left\| \frac{\pi_{\beta}^{\star}}{\pi_{\beta,\text{PRM}}^{\star}} \right\|_{\infty}\right) \tag{12}$$

What remains is to bound the worst-case density ratio between $\pi_{\beta}^{\star}$ and $\pi_{\beta,\text{PRM}}^{\star}$. Noting from eq. (11),

$$\frac{\pi_{\beta}^{\star}(y_t|x, y_{\leq t-1})}{\pi_{\beta,\text{PRM}}^{\star}(y_t|x, y_{\leq t-1})}$$

$$= \frac{e^{\beta[r_{\text{PRM}}^{\text{target}}(y_t|x, y_{\leq t-1})]}}{e^{\beta[r_{\text{PRM}}(y_t|x, y_{\leq t-1})]}} \cdot \frac{\sum_{y'} \pi_{\text{target}}(y'|x, y_{\leq t-1}) e^{\beta[r_{\text{PRM}}(y_t|x, y_{\leq t-1})]}}{\sum_{y'} \pi_{\text{target}}(y'|x, y_{\leq t-1}) e^{\beta[r_{\text{PRM}}^{\text{target}}(y_t|x, y_{\leq t-1})]}}$$

$$\leq e^{2\beta\varepsilon_{\text{PRM}}}$$

Where the last inequality uses the fact that $r_{\text{PRM}}$ satisfies Assumption 1. Combining with eq. (12) completes the proof.

## C    ANALYSIS OF SMC: PROOF OF THEOREMS 4 AND 5

We begin with the proof of Theorem 4; the last part of this section will be dedicated toward proving Theorem 5. Prior to diving into the proof, we will first introduce some additional notation. Let $N_y$ denote the number of occurrences of a particular $y \in \mathcal{Y}$ in the collection of $N \sim \mathsf{Poi}(n)_{>0}$ responses, $\mathbf{Y}_{\mathrm{draft}}$, drawn i.i.d. from $\pi_{\mathrm{gen}}(\cdot|x)$. Recall the definition of the score function $\phi_\beta(\cdot|\cdot)$ as defined in Definition 5. Furthermore, define,

$$\Phi_\beta(s) = \mathbb{E}_{Y \sim \pi_{\mathrm{gen}}(\cdot|x)} \left[ e^{-s\,\phi_\beta(Y|x)} \right]$$

as the Laplace transform of $\phi_\beta(Y|x)$ for $Y \sim \pi_{\mathrm{gen}}(\cdot|x)$. Define,

$$\pi_{\mathrm{SMC},N}(y|x) = \mathbb{E}\left[ \frac{N_y \phi_\beta(y|x)}{Z_{\mathbf{Y}_{\mathrm{draft}},\beta}(x)} \middle| N \right] \tag{13}$$

Furthermore, let us define,

$$\overline{\pi}_{\mathrm{SMC},n}(y|x) = \mathbb{E}_{N \sim \mathsf{Poi}(n)|_{>0}}[\pi_{\mathrm{SMC},N}(y|x)], \tag{14}$$

where $N_0 \sim \mathsf{Poi}(n)$. Note that with our definition of scores (Definition 5), we have,

$$\pi_\beta^\star(y|x) = \frac{\pi_{\mathrm{target}}(y|x) e^{\beta r(y|x)}}{\mathbb{E}_{Y \sim \pi_{\mathrm{target}}(\cdot|x)}[e^{\beta r(Y|x)}]} = \frac{\pi_{\mathrm{gen}}(y|x)\phi_\beta(y|x)}{\mathbb{E}_{Y \sim \pi_{\mathrm{gen}}(\cdot|x)}[\phi_\beta(Y|x)]} = \frac{\pi_{\mathrm{gen}}(y|x)\phi_\beta(y|x)}{-\Phi_\beta'(0)}.$$

Our objective is to upper bound,

$$\mathbb{E}_{N \sim \mathsf{Poi}(n)|_{>0}}\left[ D_{\chi^2}(\pi_{\mathrm{SMC},N} \| \pi_\beta^\star) \right] = \sum_{y \in \mathcal{Y}} \mathbb{E}_{N \sim \mathsf{Poi}(n)|_{>0}}\left[ \frac{(\pi_{\mathrm{SMC},N}(y|x))^2}{\pi_\beta^\star(y|x)} \right]$$

Our first result will be to provide a formula for the probability that a response $y$ is sampled by $\overline{\pi}_{\mathrm{SMC},n} = \mathbb{E}[\pi_{\mathrm{SMC},N}]$.

**Lemma 4.** $\overline{\pi}_{SMC,n}$ *can be written as the following integral,*

$$\overline{\pi}_{SMC,n}(y|x) = \frac{n\pi_{\mathrm{gen}}(y|x)\phi_\beta(y|x)}{1 - e^{-n}} \int_0^\infty e^{n(\Phi_\beta(s)-1)} \cdot e^{-s\phi_\beta(y|x)} \mathrm{d}s.$$

*Proof.* Our first step we will be to arrive at an integral representation of $1/Z_{\mathbf{Y}_{\mathrm{draft}},\beta}(x)$. For any $A > 0$, $A^{-1} = \int_0^\infty e^{-sA}\mathrm{d}s$. With the choice $A = Z_{\mathbf{Y}_{\mathrm{draft}},\beta}(x)$, by Fubini's theorem,

$$\mathbb{E}\left[ \frac{\mathbb{I}(N \geq 1) \cdot N_y \phi_\beta(y|x)}{Z_{\mathbf{Y}_{\mathrm{draft}},\beta}(x)} \right] = \phi_\beta(y|x) \int_0^\infty \mathbb{E}\left[ N_y e^{-s\,Z_{\mathbf{Y}_{\mathrm{draft}},\beta}(x)} \mathbb{I}(N \geq 1) \right] \mathrm{d}s = \int_0^\infty \Psi_y(s)\,\mathrm{d}s \tag{15}$$

where we define,

$$\Psi_y(s) = \phi_\beta(y|x)\mathbb{E}\left[ N_y e^{-s\,Z_{\mathbf{Y}_{\mathrm{draft}},\beta}(x)} \mathbb{I}(N_0 \geq 1) \right] = \phi_\beta(y|x)\mathbb{E}\left[ N_y e^{-s\,Z_{\mathbf{Y}_{\mathrm{draft}},\beta}(x)} \right],$$

which uses the fact that $N_y = 0$ if $N_0 = 0$. The remainder of this proof will be to analyze $\Psi_y(s)$. Define $N_y = \sum_{y' \in \mathbf{Y}_{\mathrm{draft}}} \mathbb{I}(y' = y)$ as the number of times $y$ appears in $\mathbf{Y}_{\mathrm{draft}}$. Then,

$$\Psi_y(s) = \mathbb{E}\left[ N_y e^{-s \sum_{y_0 \in \mathcal{Y}} N_{y_0} \phi_\beta(y_0|x)} \right]$$

$$= \mathbb{E}\left[ N_y e^{-sN_y \phi_\beta(y|x)} \right] \cdot \prod_{y_0 \in \mathcal{Y} \setminus \{y\}} \mathbb{E}\left[ e^{-sN_{y_0}\phi_\beta(y_0|x)} \right]$$

where the second equation uses independence admitted by the Poisson structure. Note that,

$$\mathbb{E}\left[ e^{-sN_{y_0}\phi_\beta(y_0|x)} \right] = \sum_{j=0}^\infty e^{-sj\phi_\beta(y_0|x)} \Pr[N_y = j]$$

$$= \sum_{j=0}^\infty e^{-sj\phi_\beta(y_0|x)} \cdot \frac{(n\pi_{\mathrm{gen}}(y_0|x))^j}{j!} e^{-n\pi_{\mathrm{gen}}(y_0|x)}$$

$$= \exp\left( n\pi_{\mathrm{gen}}(y_0|x)\left( e^{-s\phi_\beta(y_0|x)} - 1 \right) \right)$$

A similar calculation results in,

$$\mathbb{E}\left[N_y e^{-sN_y\phi_\beta(y|x)}\right] = \sum_{j=0}^{\infty} j e^{-sj\phi_\beta(y|x)} \Pr[N_y = j]$$

$$= \sum_{j=1}^{\infty} e^{-sj\phi_\beta(y|x)} \cdot \frac{(n\pi_{\text{gen}}(y|x))^j}{(j-1)!} e^{-n\pi_{\text{gen}}(y|x)}$$

$$= n\pi_{\text{gen}}(y|x) e^{-s\phi_\beta(y|x)} \sum_{j=0}^{\infty} \cdot \frac{\left(n\pi_{\text{gen}}(y|x)e^{-s\phi_\beta(y|x)}\right)^j}{j!} e^{-n\pi_{\text{gen}}(y|x)}$$

$$= n\pi_{\text{gen}}(y|x) e^{-s\phi_\beta(y|x)} \exp\left(n\pi_{\text{gen}}(y|x)\left(e^{-s\phi_\beta(y|x)} - 1\right)\right)$$

Multiplying everything out, we get,

$$\Psi_y(s) = n\pi_{\text{gen}}(y|x) e^{-s\phi_\beta(y|x)} \exp\left(n\pi_{\text{gen}}(y|x)\left(e^{-s\phi_\beta(y|x)} - 1\right)\right) \cdot \prod_{y_0 \in \mathcal{Y}\setminus\{y\}} \exp\left(n\pi_{\text{gen}}(y_0|x)\left(e^{-s\phi_\beta(y_0|x)} - 1\right)\right)$$

$$= n\pi_{\text{gen}}(y|x) e^{-s\phi_\beta(y|x)} e^{n(\Phi_\beta(s)-1)}$$

Plugging into eq. (16) completes the proof. $\qquad\square$

As a sanity check, we can check that the probability calculated in Lemma 4 integrates to 1 over $y \in \mathcal{Y}$. Notably,

$$\sum_{y\in\mathcal{Y}} \overline{\pi}_{\text{SMC},n}(y|x) = \sum_{y\in\mathcal{Y}} \frac{n\pi_{\text{gen}}(y|x)\phi_\beta(y|x)}{1 - e^{-n}} \int_0^\infty e^{n(\Phi_\beta(s)-1)} \cdot e^{-s\phi_\beta(y|x)} \mathrm{d}s$$

$$= \frac{n}{1 - e^{-n}} \int_0^\infty e^{n(\Phi_\beta(s)-1)} \cdot \mathbb{E}_{Y\sim\pi_{\text{gen}}(\cdot|x)}[\phi_\beta(Y|x)e^{-s\phi_\beta(Y|x)}] \mathrm{d}s$$

$$= \frac{-n}{1 - e^{-n}} \int_0^\infty e^{n(\Phi_\beta(s)-1)} \cdot \Phi_\beta'(s) \mathrm{d}s$$

$$\overset{(a)}{=} \frac{1}{1 - e^{-n}} \int_0^n e^{-t} \cdot \mathrm{d}t$$

$$= 1$$

where in $(a)$, $t = n(1 - \Phi_\beta(s))$.

Our next result will be to provide a formula for $\mathbb{E}[(\pi_{\text{SMC},N}(y|x))^2]$.

**Lemma 5.** $\mathbb{E}[(\pi_{\text{SMC},N}(y|x))^2]$ *has the following integral form,*

$$\mathbb{E}[(\pi_{\text{SMC},N}(y|x))^2]$$

$$= \frac{(\phi_\beta(y|x))^2}{1 - e^{-n}} \left(\int_0^\infty \int_0^\infty \left(n + n^2\Phi_\beta(s_1)\Phi_\beta(s_2)\right) \left((\pi_{\text{gen}}(y|x))^2 e^{-(s_1+s_2)\phi_\beta(y|x)}\right)\right.$$

$$\left. \times \exp\left(n\left[\Phi_\beta(s_1)\Phi_\beta(s_2) - 1\right]\right) \mathrm{d}s_1\,\mathrm{d}s_2\right)$$

*Proof.* For any $A > 0$, note that $A^{-1} = \int_0^\infty e^{-sA}\mathrm{d}s$. By Fubini's theorem, and letting $N_0 \sim \text{Poi}(n)$,

$$\underbrace{\Pr(N_0 \geq 1)}_{=1-e^{-n}} \cdot \mathbb{E}_N[(\pi_{\text{SMC},N}(y|x))^2]$$

$$= \mathbb{E}\left[\mathbb{E}\left[\frac{\mathbb{I}(N_0 \geq 1) \cdot N_y\phi_\beta(y|x)}{Z_{\mathbf{Y}_{\text{draft}},\beta}(x)}\bigg| N_0\right]^2\right]$$

$$\overset{(a)}{=} (\phi_\beta(y|x))^2 \cdot \mathbb{E}\left[\frac{\mathbb{I}(N_0 \geq 1) \cdot N_y\widetilde{N}_y}{Z_{\mathbf{Y}_{\text{draft}},\beta}(x) \cdot \widetilde{Z}_{\mathbf{Y}_{\text{draft}},\beta}(x)}\right]$$

$$= (\phi_\beta(y|x))^2 \int_0^\infty \int_0^\infty \mathbb{E}\left[N_y\widetilde{N}_y \cdot e^{-s_1 Z_{\mathbf{Y}_{\text{draft}},\beta}(x) - s_2 \widetilde{Z}_{\mathbf{Y}_{\text{draft}},\beta}(x)} \mathbb{I}(N_0 \geq 1)\right] \mathrm{d}s_1\,\mathrm{d}s_2$$

where in $(a)$, $\widetilde{N}_y$ and $\widetilde{Z}_{\mathbf{Y}_{\text{draft}}, \beta}(x)$ denote independent copies of $N_y$ and $Z_{\mathbf{Y}_{\text{draft}}, \beta}(x)$ conditioned on $N_0$. This can be further rewritten as,

$$\Pr(N_0 \geq 1) \cdot \mathbb{E}_N[(\pi_{\text{SMC}, N}(y|x))^2] = (\phi_\beta(y|x))^2 \int_0^\infty \int_0^\infty \Phi(s_1, s_2) \, \mathrm{d}s_1 \, \mathrm{d}s_2 \qquad (16)$$

where we define,

$$\Psi_y(s_1, s_2) = \mathbb{E}\left[N_y \widetilde{N}_y \cdot e^{-s_1 Z_{\mathbf{Y}_{\text{draft}}, \beta}(x) - s_2 \widetilde{Z}_{\mathbf{Y}_{\text{draft}}, \beta}(x)}\right]$$

which uses the fact that $N_y = \widetilde{N}_y = 0$ if $N_0 = 0$. The remainder of this proof will be to analyze $\Psi_y(s_1, s_2)$. First we describe a unified way to treat $\{N_y\}$ and $\{\widetilde{N}_y\}$. Across $N_0 \sim \mathsf{Poi}(n)|_{>0}$ trials, suppose we draw a $z^i = y_1^i y_2^i \sim \pi_{\text{gen}}(\cdot|x) \times \pi_{\text{gen}}(\cdot|x)$ in each round. At the end of these rounds, we will let $\sum_{i=1}^{N_0} \mathbb{I}(z_1^i = y) = N_y$ and $\sum_{i=1}^{N_0} \mathbb{I}(z_2^i = y) = \widetilde{N}_y$. For any $z \in \mathcal{Y}^2$, let $N_z$ denote the number of occurrences of $z$ among the outcomes of the trials. By the Poisson structure, we have that $N_z$ is independent across different values of $z$. Furthermore, we have that $N_y = \sum_{z \in \mathcal{Y}^2 : z_1 = y} N_z$ and $\widetilde{N}_y = \sum_{z \in \mathcal{Y}^2 : z_2 = y} N_z$. This means that,

$$\Psi_y(s_1, s_2) = \mathbb{E}\left[\left(\sum_{z : z_1 = y} N_z\right)\left(\sum_{\widetilde{z} : \widetilde{z}_2 = y} N_{\widetilde{z}}\right) \cdot e^{-\sum_{z'} N_{z'}\left(s_1 \phi_\beta(z_1'|x) + s_2 \phi_\beta(z_2'|x)\right)}\right]$$

$$= \mathbb{E}\left[\left(\sum_{\substack{z : z_1 = y \\ \widetilde{z} : \widetilde{z}_2 = y \\ z \neq \widetilde{z}}} N_z N_{\widetilde{z}} + N_{yy}^2\right) \cdot e^{-\sum_{z'} N_{z'} \alpha(z')}\right] \qquad (17)$$

Where $\alpha(z') = s_1 \phi_\beta(z_1'|x) + s_2 \phi_\beta(z_2'|x)$. Breaking down the first bracket into a summation across individual terms, we will first bound for $z \neq \widetilde{z}$ such that $z_1 = y$ and $\widetilde{z}_2 = y$,

$$\mathbb{E}\left[N_z N_{\widetilde{z}} \cdot e^{-\sum_{z'} N_{z'} \alpha(z')}\right] = \mathbb{E}\left[N_z e^{-N_z \alpha(z)}\right] \mathbb{E}\left[N_{\widetilde{z}} e^{-N_{\widetilde{z}} \alpha(\widetilde{z})}\right] \prod_{z' \notin \{z, \widetilde{z}\}} \mathbb{E}\left[e^{-N_{z'} \alpha(z')}\right] \qquad (18)$$

Let $\pi_{\text{gen}}(z'|x) = \pi_{\text{gen}}(z_1'|x)\pi_{\text{gen}}(z_2'|x)$. Then, for any $\omega \in \mathbb{R}$,

$$\mathbb{E}\left[e^{-\omega N_{z'}}\right] = \sum_{j=0}^\infty e^{-\omega j} \Pr[N_{z'} = j]$$

$$= \sum_{j=0}^\infty e^{-\omega j} \cdot \frac{\left(n\pi_{\text{gen}}(z'|x)\right)^j}{j!} e^{-n\pi_{\text{gen}}(z'|x)}$$

$$= \exp\left(n\pi_{\text{gen}}(z'|x)\left(e^{-\omega} - 1\right)\right)$$

For any $k \geq 0$, a similar calculation results in,

$$\mathbb{E}\left[e^{-\omega N_{z'}} \cdot \prod_{l=0}^k (N_{z'} - l)\right] = \sum_{j=0}^\infty \prod_{l=0}^k (j - l) e^{-\omega j} \Pr[N_{z'} = j]$$

$$= \sum_{j=k+1}^\infty e^{-\omega j} \frac{(n\pi_{\text{gen}}(z'|x))^j}{(j - k - 1)!} e^{-n\pi_{\text{gen}}(z'|x)}$$

$$= \left(n\pi_{\text{gen}}(z'|x)e^{-\omega}\right)^{k+1} \cdot \sum_{j=0}^\infty e^{-\omega j} \frac{(n\pi_{\text{gen}}(z'|x))^j}{j!} e^{-n\pi_{\text{gen}}(z'|x)}$$

$$= \left(n\pi_{\text{gen}}(z'|x)e^{-\omega}\right)^{k+1} \cdot \exp\left(n\pi_{\text{gen}}(z'|x)\left(e^{-\omega} - 1\right)\right)$$

Combining with eq. (18) (with appropriately instantiated $\omega$ and $k$),

$$\mathbb{E}\left[N_z N_{\widetilde{z}} \cdot e^{-\sum_{z'} N_{z'} \alpha(z')}\right]$$

$$= \left(n\pi_{\text{gen}}(z|x)e^{-\alpha(z)}\right)\left(n\pi_{\text{gen}}(\widetilde{z}|x)e^{-\alpha(\widetilde{z})}\right) \prod_{z'} \exp\left(n\pi_{\text{gen}}(z'|x)\left(e^{-\alpha(z')} - 1\right)\right)$$

$$= \left(n\pi_{\text{gen}}(z|x)e^{-\alpha(z)}\right)\left(n\pi_{\text{gen}}(\widetilde{z}|x)e^{-\alpha(\widetilde{z})}\right) \exp\left(n\mathbb{E}_{z' \sim \pi_{\text{gen}}(\cdot|x)}\left[e^{-s_1 \phi_\beta(z_1'|x) + s_2 \phi_\beta(z_2'|x)} - 1\right]\right)$$

$$= \left(n\pi_{\text{gen}}(z|x)e^{-\alpha(z)}\right)\left(n\pi_{\text{gen}}(\widetilde{z}|x)e^{-\alpha(\widetilde{z})}\right) \exp\left(n\left[\Phi_\beta(s_1)\Phi_\beta(s_2) - 1\right]\right)$$

On the other hand,

$$
\mathbb{E}\left[N_{yy}^2 \cdot e^{-\sum_{z'} N_{z'}\alpha(z')}\right]
$$

$$
= \mathbb{E}\left[N_{yy}^2 e^{-N_{yy}\alpha(yy)}\right] \prod_{z' \neq yy} \mathbb{E}\left[e^{-N_{z'}\alpha(z')}\right]
$$

$$
= \mathbb{E}\left[\left(N_{yy}(N_{yy}-1) + N_{yy}\right)e^{-N_{yy}\alpha(yy)}\right] \prod_{z' \neq yy} \mathbb{E}\left[e^{-N_{z'}\alpha(z')}\right]
$$

$$
= \left(\left(n\pi_{\mathrm{gen}}(yy|x)e^{-\alpha(yy)}\right)^2 + \left(n\pi_{\mathrm{gen}}(yy|x)e^{-\alpha(yy)}\right)\right) \cdot \exp\left(n\pi_{\mathrm{gen}}(z'|x)\left(e^{-\alpha(yy)}-1\right)\right)
$$

$$
\times \prod_{z' \neq yy} \exp\left(n\pi_{\mathrm{gen}}(z'|x)\left(e^{-\alpha(z')}-1\right)\right)
$$

$$
= \left(\left(n\pi_{\mathrm{gen}}(yy|x)e^{-\alpha(yy)}\right)^2 + \left(n\pi_{\mathrm{gen}}(yy|x)e^{-\alpha(yy)}\right)\right) \cdot \exp\left(n\left[\Phi_\beta(s_1)\Phi_\beta(s_2)-1\right]\right)
$$

Combining with eq. (17),

$$
\Psi_y(s_1, s_2) = \sum_{\substack{z:z_1=y \\ \widetilde{z}:\widetilde{z}_2=y}} \left(n\pi_{\mathrm{gen}}(z|x)e^{-\alpha(z)}\right)\left(n\pi_{\mathrm{gen}}(\widetilde{z}|x)e^{-\alpha(\widetilde{z})}\right) \exp\left(n\left[\Phi_\beta(s_1)\Phi_\beta(s_2)-1\right]\right)
$$

$$
+ \left(n\pi_{\mathrm{gen}}(yy|x)e^{-\alpha(yy)}\right) \cdot \exp\left(n\left[\Phi_\beta(s_1)\Phi_\beta(s_2)-1\right]\right)
$$

$$
= \Phi_\beta(s_1)\Phi_\beta(s_2)\left(n\pi_{\mathrm{gen}}(y|x)\right)^2 e^{-(s_1+s_2)\phi_\beta(y|x)} \exp\left(n\left[\Phi_\beta(s_1)\Phi_\beta(s_2)-1\right]\right)
$$

$$
+ \left(n(\pi_{\mathrm{gen}}(y|x))^2 e^{-(s_1+s_2)\phi_\beta(y|x)}\right) \cdot \exp\left(n\left[\Phi_\beta(s_1)\Phi_\beta(s_2)-1\right]\right)
$$

$$
= \left(n^2\Phi_\beta(s_1)\Phi_\beta(s_2)+n\right)\left(\pi_{\mathrm{gen}}(y|x)\right)^2 e^{-(s_1+s_2)\phi_\beta(y|x)} \exp\left(n\left[\Phi_\beta(s_1)\Phi_\beta(s_2)-1\right]\right)
$$

Plugging into eq. (16) completes the proof. $\qquad\square$

## C.1 BOUNDING THE KL-/$\chi^2$-DIVERGENCE: PROOF OF THEOREM 4

Observe that,

$$
1 + \mathbb{E}_{N\sim\mathsf{Poi}(n)|_{>0}}\left[D_{\chi^2}\left(\pi_{\mathrm{SMC},N}\|\pi_\beta^\star\right)\right] = \sum_{y\in\mathcal{Y}} \mathbb{E}_{N\sim\mathsf{Poi}(n)|_{>0}}\left[\frac{(\pi_{\mathrm{SMC},N}(y|x))^2}{\pi_\beta^\star(y|x)}\right]
$$

Recall the calculation of $\mathbb{E}[\pi_{\mathrm{SMC},N}(y|x))^2]$ in Lemma 5.
Noting that $\sum_{y\in\mathcal{Y}} \pi_{\mathrm{gen}}(y|x)\phi_\beta(y|x)e^{-(s_1+s_2)\phi_\beta(y|x)} = (-\Phi_\beta'(s_1+s_2))$,

$$
\sum_{y\in\mathcal{Y}} \mathbb{E}\left[\frac{(\pi_{\mathrm{SMC},N}(y|x))^2}{\pi_\beta^\star(y|x)}\right]
$$

$$
= \frac{1}{1-e^{-n}}\left(\int_0^\infty \int_0^\infty \left(n^2\Phi_\beta(s_1)\Phi_\beta(s_2)+n\right)\left((-\Phi_\beta'(0))(-\Phi_\beta'(s_1+s_2))\right)\right.
$$

$$
\left. \times \exp\left(n\left[\Phi_\beta(s_1)\Phi_\beta(s_2)-1\right]\right)\mathrm{d}s_1\,\mathrm{d}s_2\right) \qquad (19)
$$

Suppose that instead of the product $(-\Phi'_\beta(0))(-\Phi'_\beta(s_1 + s_2))$, we in fact had $(-\Phi'_\beta(s_1))(-\Phi'_\beta(s_2))$, we will show that the double integral exactly equals $1 - e^{-n}$. Indeed, observe that,

$$\int_0^\infty \int_0^\infty \left(n^2 \Phi_\beta(s_1)\Phi_\beta(s_2) + n\right)\left((-\Phi'_\beta(s_1))(-\Phi'_\beta(s_2))\right)\exp\left(n\left[\Phi_\beta(s_1)\Phi_\beta(s_2) - 1\right]\right)\mathrm{d}s_1\,\mathrm{d}s_2$$

$$= \int_0^\infty \int_0^\infty \left(n^2 \Phi_\beta(s_1)\Phi_\beta(s_2) + n\right)\exp\left(n\left[\Phi_\beta(s_1)\Phi_\beta(s_2) - 1\right]\right)\mathrm{d}\Phi_\beta(s_1)\,\mathrm{d}\Phi_\beta(s_2)$$

$$= \int_0^1 \int_0^1 \left(n^2 u_1 u_2 + n\right)\exp\left(n\left[u_1 u_2 - 1\right]\right)\mathrm{d}u_1\,\mathrm{d}u_2$$

$$= \int_0^1 \int_0^1 \frac{\mathrm{d}^2}{\mathrm{d}u_1 \mathrm{d}u_2}\exp\left(n\left[u_1 u_2 - 1\right]\right)\mathrm{d}u_1\,\mathrm{d}u_2$$

$$\overset{(a)}{=} g_n(1,1) - g_n(1,0) - g_n(0,1) + g_n(0,0)$$

$$= 1 - e^{-n}. \tag{20}$$

where in $(a)$, $g_n(u_1, u_2) = \exp\left(n\left[u_1 u_2 - 1\right]\right)$. In order to bound the integral in the form present in eq. (19), we split it into 2 parts; denoting $p_n = \log\left(n^4\right)/n$,

$$\mathcal{R}_0 = \{(s_1, s_2) \in \mathbb{R}^2 : \min\{\Phi_\beta(s_1), \Phi_\beta(s_2)\} \geq 1 - p_n\}$$

$$\mathcal{R}_1 = \{(s_1, s_2) \in \mathbb{R}^2 : \min\{\Phi_\beta(s_1), \Phi_\beta(s_2)\} \leq 1 - p_n\}.$$

And let $\mathcal{I}_0$ and $\mathcal{I}_1$ denote the integral eq. (19) (which is over $\mathbb{R}^2$) over the disjoint regions $\mathcal{R}_0$ and $\mathcal{R}_1$.

**Lemma 6** (Bounding the integral $\mathcal{I}_0$). *Assuming that $n$ is sufficiently large so that $0 \leq p_n \leq \frac{1}{\sqrt{2}}$,*

$$\mathcal{I}_0 \leq \exp\left(p_n^2\left(\phi_{\max}/\phi_{\min} - 1\right)^2\right).$$

*Recall here, that $\phi_{\max} \triangleq \max_{x \in \mathcal{X}, y \in \mathcal{Y}} \phi_\beta(y|x)$ and $\phi_{\min} \triangleq \min_{x \in \mathcal{X}, y \in \mathcal{Y}} \phi_\beta(y|x)$.*

*Proof.* Observe that $\mathcal{I}_0$ corresponds to integrating around a small neighborhood of $(0, 0)$, and to this end, we first bound $(-\Phi'_\beta(0))(-\Phi'_\beta(s_1 + s_2))$ for $(s_1, s_2) \in \mathcal{R}_0$, showing that it is approximately equal to $(-\Phi'_\beta(s_1))(-\Phi'_\beta(s_2))$ in this regime. The resulting integral can be bounded using the calculation done in eq. (20).

As $s_1$ and $s_2$ become smaller, which is the case as $n$ grows, the approximation becomes better: this essentially follows from the fact that $f(s) = \log\left(-\Phi'_\beta(s)\right)$ is a bounded, smooth convex function and behaves locally linearly. Formally,

$$f(s_1 + s_2) + f(0) - f(s_1) - f(s_2)$$

$$= \int_0^{s_1} \int_0^{s_2} f''(u_1 + u_2)\mathrm{d}u_1\mathrm{d}u_2$$

$$\overset{(a)}{=} \int_0^{s_1} \int_0^{s_2} \frac{\Phi'''_\beta(u)\Phi'_\beta(u) - (\Phi''_\beta(u))^2}{(\Phi'_\beta(u))^2}\mathrm{d}u_1\mathrm{d}u_2$$

$$= \int_0^{s_1} \int_0^{s_2} \frac{\Phi'''_\beta(u)\Phi'_\beta(u) - (\Phi''_\beta(u))^2}{(\Phi'_\beta(u))^2\Phi'_\beta(u_1)\Phi'_\beta(u_2)}\mathrm{d}\Phi_\beta(u_1)\mathrm{d}\Phi_\beta(u_2)$$

$$\leq \max_{u_1 \in [0, s_1]}\max_{u_2 \in [0, s_2]}\frac{\Phi'''_\beta(u)\Phi'_\beta(u) - (\Phi''_\beta(u))^2}{(\Phi'_\beta(u))^2\Phi'_\beta(u_1)\Phi'_\beta(u_2)} \cdot \int_0^{s_1} \int_0^{s_2} \mathrm{d}\Phi_\beta(u_1)\mathrm{d}\Phi_\beta(u_2)$$

$$\leq p_n^2 \cdot \max_{u_1 \in [0, s_1]}\max_{u_2 \in [0, s_2]}\frac{\Phi'''_\beta(u)\Phi'_\beta(u) - (\Phi''_\beta(u))^2}{(\Phi'_\beta(u))^2\Phi'_\beta(u_1)\Phi'_\beta(u_2)} \tag{21}$$

where in $(a)$, $u = u_1 + u_2$. In the $\mathcal{R}_0$ regime, letting $Y, Y' \overset{\text{i.i.d.}}{\sim} \pi_{\text{gen}}(\cdot|x)$,

$$\max_{u_1 \in [0, s_1]} \max_{u_2 \in [0, s_2]} \frac{\Phi_\beta'''(u)\Phi_\beta'(u) - (\Phi_\beta''(u))^2}{(\Phi_\beta'(u)^2 \Phi_\beta'(u_1)\Phi_\beta'(u_2)}$$

$$= \frac{\mathbb{E}\big[\phi_\beta(Y|x)\phi_\beta(Y'|x) \cdot (\phi_\beta(Y|x) - \phi_\beta(Y'|x))^2 \cdot e^{-u(\phi_\beta(Y|x)+\phi_\beta(Y'|x))}\big]}{2\mathbb{E}\big[\phi_\beta(Y|x) \cdot e^{-u(\phi_\beta(Y|x)}\big]^2 \cdot \mathbb{E}\big[\phi_\beta(Y|x)\phi_\beta(Y'|x) \cdot e^{-u(\phi_\beta(Y|x)+\phi_\beta(Y'|x))}\big]}$$

$$\overset{(a)}{\leq} \frac{\mathbb{E}\big[\phi_\beta(Y|x)\phi_\beta(Y'|x) \cdot (\phi_\beta(Y|x) - \phi_\beta(Y'|x))^2 \cdot e^{-u(\phi_\beta(Y|x)+\phi_\beta(Y'|x))}\big]}{2\phi_{\min}^2(1 - p_n)^2 \cdot \mathbb{E}\big[\phi_\beta(Y|x)\phi_\beta(Y'|x) \cdot e^{-u(\phi_\beta(Y|x)+\phi_\beta(Y'|x))}\big]}$$

$$\leq \frac{(\phi_{\max} - \phi_{\min})^2}{2\phi_{\min}^2(1 - p_n)^2}$$

$$\leq (\phi_{\max}/\phi_{\min} - 1)^2,$$

where $(a)$ uses the assumption that $\Phi_\beta(u_1), \Phi_\beta(u_2) \geq 1 - p_n$ in this regime, and the last inequality assumes that $n$ is sufficiently large that $p_n \leq \frac{1}{\sqrt{2}}$. Combining with eq. (21),

$$\frac{(-\Phi_\beta'(s_1 + s_2))(-\Phi_\beta'(0))}{(-\Phi_\beta'(s_1))(-\Phi_\beta'(s_2))} \leq \exp\left(p_n^2\left(\frac{\phi_{\max}}{\phi_{\min}} - 1\right)^2\right).$$

Combining with eq. (20) completes the proof. $\qquad\square$

**Lemma 7** (Bounding the integral $\mathcal{I}_1$). *We have that,*

$$\mathcal{I}_1 \leq \frac{4}{n^2} \cdot \frac{\phi_{\max}}{\phi_{\min}}$$

*Proof.* In the $\mathcal{R}_1$ regime, we have that $n(1 - \Phi_\beta(s_1)\Phi_\beta(s_2)) \geq \log(n^4)$. Therefore,

$$\mathcal{I}_1 = \frac{1}{1 - e^{-n}}\left(\int_{\mathcal{R}_1} \left(n^2 \Phi_\beta(s_1)\Phi_\beta(s_2) + n\right)\left((-\Phi_\beta'(0))(-\Phi_\beta'(s_1 + s_2))\right)\right.$$

$$\left. \exp\big(n\big[\Phi_\beta(s_1)\Phi_\beta(s_2) - 1\big]\big)\mathrm{d}s_1\,\mathrm{d}s_2\right)$$

$$\leq 4n^2 \int_{\mathcal{R}_1} \left((-\Phi_\beta'(0))(-\Phi_\beta'(s_1 + s_2))\right)\exp\big(-\log(n^4)\big)\mathrm{d}s_1\,\mathrm{d}s_2$$

$$= \frac{4}{n^2}\int_{\mathcal{R}_1} \mathbb{E}[\phi_\beta(Y|x)] \cdot \mathbb{E}\big[\phi_\beta(Y|x)e^{-(s_1+s_2)\phi_\beta(Y|x)}\big]\mathrm{d}s_1\,\mathrm{d}s_2$$

$$= \frac{4}{n^2} \cdot \mathbb{E}[\phi_\beta(Y|x)] \cdot \mathbb{E}\big[\phi_\beta(Y|x))^{-1}\big]$$

$$= \frac{4}{n^2} \cdot \frac{\phi_{\max}}{\phi_{\min}}$$

$$\leq \frac{4}{n^2} \cdot \frac{\phi_{\max}}{\phi_{\min}}$$

$\qquad\square$

## C.2 PROOF OF THEOREM 4

*Proof.* This is a consequence of the fact that by eq. (19),

$$1 + D_{\chi^2}(\overline{\pi}_{\text{SMC},n}\|\pi_\beta^\star) = \mathcal{I}_0 + \mathcal{I}_1$$

Combining with Lemmas 6 and 7, this results in the bound,

$$1 + \mathbb{E}[D_{\chi^2}(\pi_{\text{SMC},N}\|\pi_\beta^\star)] \leq \frac{\phi_{\max}}{\phi_{\min}} \cdot \frac{4}{n^2} + \exp\left(p_n^2\left(\frac{\phi_{\max}}{\phi_{\min}} - 1\right)^2\right).$$

Simplifying this by noting that $z \le e^z - 1$, gives the inequality,

$$1 + \mathbb{E}[D_{\chi^2}(\pi_{\mathrm{SMC},N}\|\pi_\beta^\star)] \le 1 + 2\left(\exp\left(4p_n^2\left(\frac{\phi_{\max}}{\phi_{\min}}\right)^2\right) - 1\right).$$

Taking a logarithm on both sides, noting that $D_{\mathrm{KL}}(p\|q) \le \log(1 + D_{\chi^2}(p\|q))$ and simplifying the RHS with the inequality, $\log(1 + c(e^x - 1)) \le cx$ for $c \ge 1$ completes the proof. $\qquad\square$

### C.3 PROOF OF THEOREM 5

Consider the following problem instance where there is a single prompt (which we will avoid stating in the reward and policy notations), and the response space $\mathcal{Y} = \{y_0, y_1\}$. $r(y_0) = 0$ and $r(y_1) = R$, and $1 - \pi_{\mathrm{target}}(y_0) = \pi_{\mathrm{target}}(y_1) = \frac{1}{1+\theta}$ where we denote $\theta = e^{\beta R}$ for the sake of brevity. The optimal aligned model is,

$$\pi_{w,\beta}^\star(y_0) = \pi_{w,\beta}^\star(y_1) = \frac{1}{2}$$

For this model, the Laplace transform of the score function is,

$$\Phi_\beta(s) = \mathbb{E}_{Y\sim\pi_{\mathrm{target}}(\cdot|x)}[e^{-s\phi_\beta(Y|x)}] = \frac{\theta e^{-s} + e^{-s\theta}}{1+\theta} \qquad (22)$$

From Lemma 5,

$$\overline{\pi}_{\mathrm{SMC},n}(y_0) = \frac{\theta}{1+\theta} \cdot \frac{n}{(1-e^{-n})} \int_0^\infty e^{n(\Phi_\beta(s)-1)} \cdot e^{-s}\mathrm{d}s$$

$$= \frac{\theta}{1+\theta} \cdot \frac{n}{(1-e^{-n})} \int_0^1 e^{n\left(\frac{\theta t + t^\theta}{1+\theta}-1\right)}\mathrm{d}t$$

where the last equation uses the structure of the Laplace transform of the score function in eq. (22). Let $h = g(t) = \frac{\theta t + t^\theta}{1+\theta}$, and thereby, $t = g^{-1}(h)$ and $\mathrm{d}t = 1/g'(g^{-1}(h))\mathrm{d}h$.

$$\overline{\pi}_{\mathrm{SMC},n}(y_0) = \frac{\theta}{1+\theta} \cdot \frac{n}{(1-e^{-n})} \int_0^1 \frac{e^{n(h-1)}}{g'(g^{-1}(h))}\mathrm{d}h$$

$$\stackrel{(b)}{=} \frac{\theta}{1+\theta} \cdot \frac{1}{(1-e^{-n})} \int_0^n \frac{e^{-h_1}}{g'(g^{-1}(1-h_1/n))}\mathrm{d}h_1$$

where in $(b)$ we plug in $h_1 = n(1-h)$. Observe that $g'(t) = \frac{\theta + \theta t^{\theta-1}}{1+\theta} \le \frac{2\theta}{1+\theta}$ is an increasing function in $t$, and thereby,

$$\overline{\pi}_{\mathrm{SMC},n}(y_0) \ge \frac{1}{2}.$$

In particular, we will show that $\left|\overline{\pi}_{\mathrm{SMC},n}(y_0) - \frac{1}{2}\right| \ge \mathcal{O}(\frac{\theta}{n})$. Since $g'$ and $g^{-1}$ are increasing functions,

$$\overline{\pi}_{\mathrm{SMC},n}(y_0) \ge \frac{\theta}{1+\theta} \cdot \frac{1}{(1-e^{-n})} \left[\int_0^1 \frac{e^{-h_1}}{g'(g^{-1}(1))}\mathrm{d}h_1 + \int_1^n \frac{e^{-h_1}}{g'(g^{-1}(1-1/n))}\mathrm{d}h_1\right]$$

$$= \frac{\theta}{1+\theta} \cdot \frac{1}{(1-e^{-n})} \left[(1-e^{-1})\frac{1+\theta}{2\theta} + \frac{e^{-1}-e^{-n}}{2\theta/(1+\theta)} \cdot \frac{2\theta/(1+\theta)}{g'(g^{-1}(1-1/n))}\right]$$

$$= \frac{1}{2} + \frac{e^{-1}-e^{-n}}{2}\left(\frac{2\theta/(1+\theta)}{g'(g^{-1}(1-1/n))} - 1\right) \qquad (23)$$

Observe that,

$$g\left(1 - \frac{1+\theta}{2\theta n}\right) = \frac{\theta}{1+\theta}\left(1 - \frac{1+\theta}{2\theta n}\right) + \frac{1}{1+\theta}\left(1 - \frac{1+\theta}{2\theta n}\right)^\theta$$

$$\ge \frac{\theta}{1+\theta}\left(1 - \frac{1+\theta}{2\theta n}\right) + \frac{1}{1+\theta}\left(1 - \frac{1+\theta}{2n}\right)$$

$$= 1 - \frac{1}{n}$$

Since $g$ (and thereby $g^{-1}$) is increasing, this results in the inequality, $g^{-1}(1 - 1/n) \leq 1 - \frac{1+\theta}{2\theta n}$. Since $g'$ is also increasing, we have the inequality,

$$g'(g^{-1}(1 - 1/n)) \leq g'\left(1 - \frac{1+\theta}{2\theta n}\right)$$

$$= \frac{\theta}{1+\theta} + \frac{\theta}{1+\theta}\left(1 - \frac{1+\theta}{2\theta n}\right)^{\theta-1}$$

$$\leq \frac{2\theta}{1+\theta} - \frac{\theta}{1+\theta}\left[1 - \left(1 - \frac{1+\theta}{2\theta n}\right)^{\theta-1}\right]$$

Assuming that $n \geq \theta \geq 2$, $\left(1 - \frac{1+\theta}{2\theta n}\right)^{\theta-1} \geq 1 - \frac{1+\theta}{4n}$. This implies,

$$g'(g^{-1}(1 - 1/n)) \leq \frac{2\theta}{1+\theta} - \frac{\theta}{4n}.$$

Combining with eq. (23),

$$\overline{\pi}_{\text{SMC},n}(y_0) - \frac{1}{2} \geq \frac{e^{-1} - e^{-n}}{2}\left(\frac{2\theta/(1+\theta)}{2\theta/(1+\theta) - \theta/4n} - 1\right)$$

$$\geq \frac{e^{-1} - e^{-n}}{2}\left(\frac{2}{2 - \theta/4n} - 1\right)$$

$$\geq \frac{\theta}{50n},$$

where the last inequality assumes that $n \geq 3$. This implies that,

$$D_{\text{TV}}(\overline{\pi}_{\text{SMC},n}, \pi_\beta^\star) \geq \frac{\theta}{50n}.$$

The proof concludes by an application of Pinsker's inequality to lower bound the KL divergence by the squared TV distance, $D_{\text{TV}}^2(p, q) \leq \frac{1}{2}D_{\text{KL}}(p, q)$). Finally, by the convexity of the KL divergence, we may take the expectation over $N \sim \text{Poi}(n)|_{>0}$ outside the KL divergence.

# D  ADDITIONAL EXPERIMENTAL DETAILS

## D.1  LATENCY BREAKDOWN

In this experiment, we further hope to break down and understand the distribution of latency cost across the various components of SPECS: generation from the draft model (when deferred to), generation from the target model, and scoring using the draft and target models. We evaluate these across all three datasets, AMC23, MATH500 and OlympiadBench when the draft-target pair is Qwen2.5-1.5B-Instruct and Qwen2.5-7B-Instruct. We observe that generation from the target model is still a large component of the latency breakdown, since it is slower than the other steps. However, the overall latency can be brought down via generation from the draft model.

## D.2  HARDWARE AND FRAMEWORKS USED FOR THE EXPERIMENTS

All experiments are performed on a node equipped with 3 NVIDIA A100 Tensor Core GPUs. Each model is served on a separate GPU and accessed using vLLM (Kwon et al., 2023) API endpoint. For both beam search and SPECS, we vary the computational budget (e.g., by adjusting the beam width $n$) to trace out different latency-accuracy scalings.

## D.3  ENGINEERING AND PERFORMANCE OPTIMIZATIONS:

In order to make the SPECS algorithm more latency-efficient, we implemented some performance optimizations for increased parallelism and efficiency that are worth noting here. In particular, within the SUBSAMPLE subroutine in Algorithm 2, the log-likelihoods of the generated beams under the target model as well as their scores under the reward model are evaluated. Both of these options can

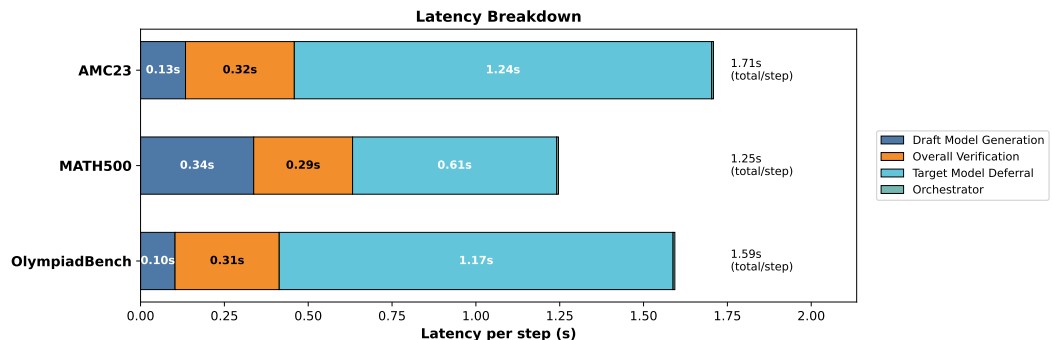

Figure 5: Latency breakdown is shown

be carried out concurrently to only pay for the latency cost of the slower of these two operations. Moreover, during the generation of new candidate beams for a given step, ideally we would like to use prefix-caching so that we do not need to recompute the logits of the previously decoded steps. However, prefix-caching is disabled in vLLM when we also want to access the log probabilities of the prompt tokens, which is necessary for the SUBSAMPLE subroutine that requires the computation of log-likelihoods of trajectories under the target model. We solve this problem by changing the vLLM source code to allow the computation of log probabilities. The solution to this problem is that we do not actually need to compute the log-probabilities of the tokens that are prefix-cache hit, we only need the log-probabilities of the tokens appearing in the last step. We solve this issue by bypassing the prefix-cache hit tokens for the previous steps and efficiently compute the log-probabilities of the newly generated tokens in the last step, avoiding redundant computations.

### D.4 HYPERPARAMETERS FOR SPECS EXPERIMENTS

The hyperparameters used in the experiments in Table 1 are presented in Table 3. Importantly, there are 3 key hyperparameters in this algorithm: $n$ is the beam-width, $\beta$ is the inverse-temperature for the soft verification, $\tau$ is the dynamic switching threshold. For MATH500 and OlympiadBench datasets, the hyperparameters are chosen on a separate set of 100 questions from that dataset, treated as a validation set. For AMC23 dataset, since there were only 40 questions, we select the hyperparameters that give the best accuracy vs. latency for different choices of $n$.

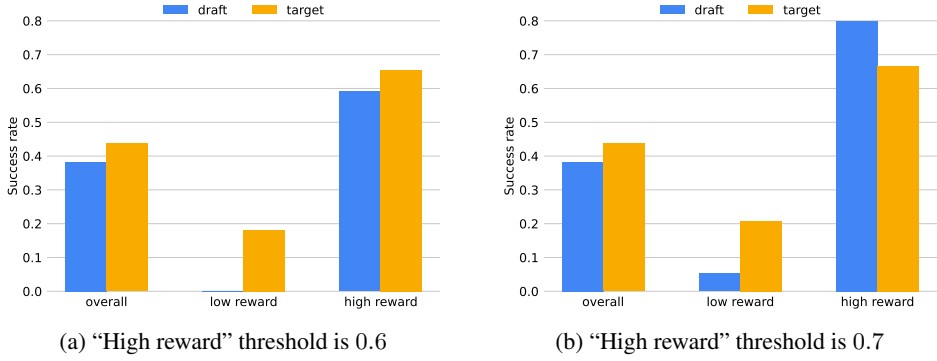

(a) "High reward" threshold is 0.6       (b) "High reward" threshold is 0.7

Figure 6: The threshold on reward determining the batch of "high-reward" 8-step partial traces generated by the target model is varied, within the same experimental setup of Figure 2b. The trend persists very strongly: the draft model is much better at completing high-reward partial traces compared to low-reward partial traces generated by the target model.

Table 3: Chosen hyperparameters $\beta$ and $\tau$ for different datasets; **n** is the beam width.

| Family | Dataset | n | $\beta$ | $\tau$ |
|---|---|---|---|---|
| Qwen Family | AMC23 | 4 | 20 | 0.8 |
| | | 8 | 16 | 0.8 |
| | | 16 | 12 | 0.9 |
| | MATH500 | 4 | 1000 | 0.8 |
| | | 8 | 10000 | 0.75 |
| | | 16 | 1000 | 0.8 |
| | OlympiadBench | 4 | 12 | 0.9 |
| | | 8 | 8 | 0.9 |
| | | 16 | 12 | 0.7 |
| Llama Family | AMC23 | 4 | 1000 | 0.9 |
| | | 8 | 1000 | 0.8 |
| | | 16 | 1000 | 0.85 |
| | MATH500 | 4 | 100 | 0.8 |
| | | 8 | 1000 | 0.8 |
| | | 16 | 100 | 0.7 |
| | OlympiadBench | 4 | 10000 | 0.8 |
| | | 8 | 100 | 0.6 |
| | | 16 | 1000 | 0.7 |

## D.5 DYNAMIC SWITCHING

In this section, we include some further analyses of how the accuracy varies as the reward threshold which defines "high-reward" and "low-reward" is varied in the setting of Figure 2b, where it is set as 0.5. As we increase the reward threshold defining high-reward, we observe that the trend grows stronger: the draft model gets increasingly better at solving high-reward traces. The results are described in Figure 6.

We further try to understand the effect of switching from target model as generator to draft model as generator, as a function of the point at which traces are sliced. In Figure 2b we observed that the draft model is very successful at completing 8-step high-reward partial traces generated by the target model, while it is much worse at completing the corresponding low-reward partial traces. In this section, we run the same experiment but set the switchpoint to be earlier, by generating only 4 initial steps of reasoning from the target model instead of 8. The results of this experiment are plotted in Figure 7. We observe that the draft model continues to achieve higher accuracy when completing high-reward traces compared to when completing low-reward traces. However, when we set the switchpoint to be earlier, we observe that completing with either draft or target model achieves similar performance on the low-reward traces, while there is a deterioration in performance on high-reward traces when the draft model is used to complete them.

Based on this experiment, we conclude that the trend observed in Figure 2b is stronger when we switch from the target to the draft model at later points: completing high-reward traces using the draft model is better than using the target model, when the high-rewards are observed later in the trace.

## D.6 PERFORMANCE OF SPECS UNDER NOISY PRM

We also conduct some experiments where the quality of the PRM is worsened by adding explicit Gaussian noise. In the particular, the PRM we consider is $\tilde{r}_{\mathrm{PRM}} = r_{\mathrm{PRM}} + Z$, where $Z$ is independent 0-mean Gaussian noise with standard deviation 0.1, i.e. $Z \sim N(0, 0.01)$. In this regime, we see that the performance gap between SPECS and beam search with small model increases, while the performance of SPECS still remains competitive with beam search with the large model. We also do not carry out any additional hyperparameter search prior to running SPECS in this setting, showing the robustness of our hyperparameters. The results are presented in the Table 4.

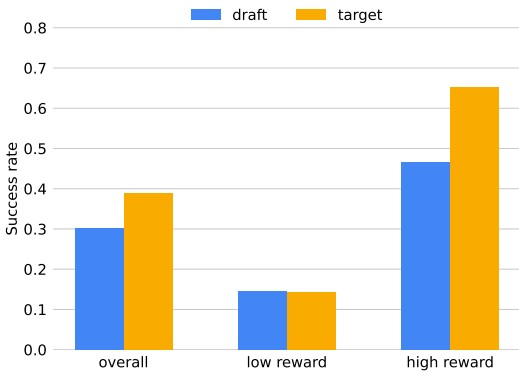

Figure 7: We generate $4$ steps of reasoning from the target model. The remaining steps are completed either by the target model, or by the draft model. We bucket the initial $4$-step reasoning traces generated by the target model into "low-reward" and "high-reward" based on the PRM reward of the trace, where the threshold is set as reward $= 0.5$. When contrasted with Figure 2b, which is the same experiment, but with $8$ initial steps of reasoning generated by the target model, there are some differences.

Table 4: Performance comparison of different methods under a noisy version of the PRM, $\tilde{r} = r + Z$, where $Z \sim N(0, 0.01)$. Here $n$ denotes the per-step beam width.

| Method | $n$ | Success Rate (%) ↑ | Runtime (s) ↓ | Success Rate (%) ↑ | Runtime (s) ↓ | Success Rate (%) ↑ | Runtime (s) ↓ |
|---|---|---|---|---|---|---|---|
| | | AMC23 | | MATH500 | | OlympiadBench | |
| BS(S) (with $\tilde{r}$) | 4 | 36.25 | 9.55 | 53.33 | 7.07 | 26.00 | 11.38 |
| BS(B) (with $\tilde{r}$) | 4 | 43.33 | 14.44 | 76.67 | 11.18 | 30.67 | 14.38 |
| SPECS (with $\tilde{r}$) | 4 | 46.67 | 16.50 | 78.00 | 11.07 | 30.67 | 14.10 |
| BS(S) (with $\tilde{r}$) | 8 | 34.38 | 12.07 | 62.00 | 9.77 | 25.00 | 15.83 |
| BS(B) (with $\tilde{r}$) | 8 | 50.83 | 20.59 | 78.67 | 14.21 | 34.00 | 20.29 |
| SPECS (with $\tilde{r}$) | 8 | 50.83 | 19.64 | 71.33 | 12.49 | 35.33 | 19.99 |
| BS(S) (with $\tilde{r}$) | 16 | 43.13 | 19.73 | 59.33 | 15.32 | 29.33 | 25.38 |
| BS(B) (with $\tilde{r}$) | 16 | 46.67 | 26.62 | 73.33 | 18.57 | 32.67 | 25.49 |
| SPECS (with $\tilde{r}$) | 16 | 49.17 | 27.57 | 72.67 | 18.76 | 37.00 | 27.99 |

### D.7 PERFORMANCE OF SPECS UNDER CONSTRAINT VALUE OF HYPERPARAMETERS

We see that selecting $\beta = 2 \times L$ where $L$ is the max token budget for a generation of a beam search step is a stable choice for $\beta$ across different benchmarks and different beam width sizes. Indeed, to demonstrate the robustness of our method to hyperparameters, we show that most of the performance of our method can be recovered by restricting the space of parameter search to a restricted subset. In particular, we set $\beta = \beta_0 * L$ where $\beta_0 = 2$ and $L$ is the token budget. Further constraining $\tau$ to be values in $tau \in \{0.7, 0.8, 0.9\}$, we report the accuracy and latency numbers of restricted search compared to an exhaustive search over $\beta$ and $\tau$ in Table 5.

Table 5: We can achieve similar performance with SPECS even if we constraint the hyperparameters to specific values across benchmarks and beam width sizes.

| Method | $n$ | Success Rate (%) ↑ | Runtime (s) ↓ | Success Rate (%) ↑ | Runtime (s) ↓ | Success Rate (%) ↑ | Runtime (s) ↓ |
|---|---|---|---|---|---|---|---|
| | | AMC23 | | MATH500 | | OlympiadBench | |
| SPECS | 4 | 59.2 | 14.8 | 77.0 | 9.83 | 41.3 | 13.1 |
| SPECS (with const. $\beta, \tau$) | 4 | 53.1 | 13.3 | 77.0 | 9.80 | 38.5 | 12.5 |
| SPECS | 8 | 62.5 | 20.7 | 79.5 | 11.3 | 41.7 | 16.9 |
| SPECS (with const. $\beta, \tau$) | 8 | 58.4 | 20.2 | 75.3 | 11.0 | 41.0 | 18.1 |
| SPECS | 16 | 68.3 | 28.8 | 81.0 | 15.8 | 44.0 | 24.6 |
| SPECS (with const. $\beta, \tau$) | 16 | 64.8 | 26.4 | 81.0 | 15.8 | 42.6 | 23.8 |

## D.8 Further ablations about SPECS

We also ran further ablations about SPECS. The main objective of these ablations is to strengthen the benefit of our proposed scoring mechanism with the loglikelihood term added, as opposed to just using the PRM rewards.

First, we show that the version of SPECS where we start with the small model first and switch to the larger model if none of the small model generations exceed the reward thesthold $\tau$ performs better than RSD, which also starts with a small model and then switches to the big model if small model generations don't meed a PRM reward threshold but chooses the beams from the small model with hard best of n sampling. Second, we show SPECS performs better than the version of RSD where the initial step is done by the large model. Thirdly, we show that the case of SPECS with no log-likelihood term in the scoring function performs worse than the proposed SPECS, showing the usefulness of the scoring function. We are using Qwen2.5-Instruct family of models here.

Table 6: Performance of different case studies with SPECS and RSD

| Method | $n$ | Success Rate (%) ↑ | Runtime (s) ↓ | Success Rate (%) ↑ | Runtime (s) ↓ | Success Rate (%) ↑ | Runtime (s) ↓ |
|---|---|---|---|---|---|---|---|
| | | AMC23 | | MATH500 | | OlympiadBench | |
| SPECS | 4 | 59.2 | 14.8 | 77.0 | 9.83 | 41.3 | 13.1 |
| RSD | 4 | 31.2 | 12.6 | 77.8 | 10.1 | 33.0 | 17.3 |
| SPECS (start with small model) | 4 | 58.9 | 18.5 | 77.5 | 13.0 | 41.0 | 15.3 |
| RSD (start with large model) | 4 | 43.3 | 16.8 | 70.0 | 13.8 | 28.0 | 23.3 |
| SPECS (with no log-likelihood term in score) | 4 | 43.2 | 12.5 | 71.5 | 8.73 | 32.5 | 16.6 |
| SPECS | 8 | 62.5 | 20.7 | 79.5 | 11.3 | 41.7 | 16.9 |
| RSD | 8 | 49.2 | 20.3 | 78.3 | 11.8 | 33.3 | 21.7 |
| SPECS (start with small model) | 8 | 61.4 | 21.9 | 78.9 | 12.7 | 41.3 | 18.1 |
| RSD (start with large model) | 8 | 39.2 | 21.2 | 66.7 | 17.8 | 32.7 | 26.0 |
| SPECS (with no log-likelihood term in score) | 8 | 47.5 | 17.9 | 73.3 | 10.6 | 32.8 | 23.8 |
| SPECS | 16 | 68.3 | 28.8 | 81.0 | 15.8 | 44.0 | 24.6 |
| RSD | 16 | 50.0 | 25.0 | 79.0 | 16.3 | 33.5 | 28.7 |
| SPECS (start with small model) | 16 | 67.9 | 29.2 | 81.0 | 17.2 | 44.2 | 27.8 |
| RSD (start with large model) | 16 | 48.3 | 26.2 | 71.3 | 22.1 | 30.7 | 35.4 |
| SPECS (with no log-likelihood term in score) | 16 | 48.6 | 28.1 | 74.5 | 14.5 | 33.1 | 27.1 |

## D.9 GPQA Experiments

Using the Qwen family of models we have in this paper, we obtain more results on the GPQA dataset. The results are presented below in Table 7. Note that SPECS achieves much better latency compared to the Beam Search with Large Model while achieving similar accuracy results, demonstrating the applicability of SPECS to reasoning problems other than mathematics:

Table 7: SPECS results on the GPQA benchmark. Here $n$ denotes the per-step beam width.

| Dataset | $n$ | SPECS | | Beam Search (Big Model) | | Beam Search (Small Model) | |
|---|---|---|---|---|---|---|---|
| | | Acc (%) ↑ | Latency (s) ↓ | Acc (%) ↑ | Latency (s) ↓ | Acc (%) ↑ | Latency (s) ↓ |
| GPQA | 4 | 30.8 | 9.00 | 34.2 | 18.6 | 17.8 | 7.77 |
| GPQA | 8 | 32.2 | 21.83 | 32.5 | 23.1 | 22.6 | 10.31 |
| GPQA | 16 | 34.5 | 22.42 | 33.3 | 28.3 | 24.2 | 15.08 |

