# OpenReview forum: "SPECS: Faster Test-Time Scaling through Speculative Drafts and Dynamic Switching"
_ICLR.cc/2026/Conference — Submitted to ICLR 2026_

### Official Review · Reviewer_pgD8 · 2025-10-26

**Soundness:** 3
**Presentation:** 4
**Contribution:** 2
**Rating:** 4
**Confidence:** 5

**Summary:**

This paper proposes SPECS, a new test-time scaling algorithm for LLMs.
SPECS combines best-of-n sampling with guidance from a process reward model (PRM) alongside a few other ideas: using a draft model (similar to speculative decoding) to accelerate inference; *tilting* the reward by the log-density ratio between the target and draft model; and adding dynamic switching between generation from the draft and target model (depending on whether the problem is deemed difficult or easy, based on the rewards).
The authors provide an extensive theoretical framework for their algorithm, although much of the theory does not seem to be applicable to the algorithm in practice (see below). The authors validate the proposed algorithm with experiments on three datasets (AMC23, MATH500, OlympiadBench), which show that it performs better than beam search with the draft model and sometimes even with the target model, and reward-guided speculative decoding (RSD) [1], a recently proposed alternative algorithm with many similarities to SPECS.

As outlined below, I believe SPECS proposes some interesting algorithmic ideas, but the novelty of the contribution is limited, and the supporting theory is not directly applicable as it hinges on assumptions that do not hold in practice. **I lean towards rejection of the paper, but believe it could also be accepted.**

[1] Baohao Liao, Yuhui Xu, Hanze Dong, Junnan Li, Christof Monz, Silvio Savarese, Doyen Sahoo, and Caiming Xiong. Reward-Guided Speculative Decoding for Efficient LLM Reasoning. 2025. URL: https://arxiv.org/abs/2501.19324.

**Strengths:**

### Algorithm
The proposed algorithm is interesting and, as the authors highlight in the Appendix, unlike other methods from the literature like RSD [1], SPECS *starts* by decoding from the target model, and only falls back to the draft model if deemed advantageous (i.e., when the reward is high). This idea is novel and nice.

### Theory
The authors prove that under certain assumptions, SPECS converges to the optimal tilted policy as $n\to\infty$. However, these assumptions do not hold in practice (see below).

### Presentation
The paper is well written, structured, and easy to read.

### Experiments
The authors provide a reasonable amount of experiments to validate the performance of SPECS (although more datasets never hurt!). The results include confidence intervals (although as far as I see, the authors do not specify how these are computed, i.e. over how many runs and what degree of confidence). Furthermore, the authors provide ablations over the latency and the effect of the dynamic switching, which show that dynamically switching based on the rewards is better than switching randomly between target and draft model (with identical proportions of models). The Appendix contains additional experiments.


[1] Baohao Liao, Yuhui Xu, Hanze Dong, Junnan Li, Christof Monz, Silvio Savarese, Doyen Sahoo, and Caiming Xiong. Reward-Guided Speculative Decoding for Efficient LLM Reasoning. 2025. URL: https://arxiv.org/abs/2501.19324.

**Weaknesses:**

### Contribution
The contribution seems relatively limited to me. Large parts of the method have already been proposed in previous works: RSD [1] proposes combining a draft model with a reward model for stepwise speculative generation (although only for $n=1$). GSI [2] proposes using the same tilted rewards as SPECS. The main novelty in SPECS seems to be that generation *starts* with the target model, and can *dynamically switch* between draft and target model.

### Theory
Theorem 1 shows that SPECS can approximate the optimal tilted distribution. However, this theorem hinges on two crucial assumptions, neither of which holds in practice: (1) It assumes that the given PRM is "idealized", meaning it corresponds to the true value function of the  policy w.r.t. a reward model (Definition 1). (2) The number of drafts generated (which is fixed in practice) is assumed to be Poisson distributed.

Furthermore, it is not clear to me what "for reasoning problems over H blocks" means exactly. Does this assume that all steps have the same length? Also, I have not checked the proofs carefully, as they are extremely extensive.


[1] Baohao Liao, Yuhui Xu, Hanze Dong, Junnan Li, Christof Monz, Silvio Savarese, Doyen Sahoo, and Caiming Xiong. Reward-Guided Speculative Decoding for Efficient LLM Reasoning. 2025. URL: https://arxiv.org/abs/2501.19324.

[2] Jonathan Geuter, Youssef Mroueh, and David Alvarez-Melis. Guided Speculative Inference for Efficient Test-Time Alignment of LLMs. 2025. URL: https://arxiv.org/abs/2506.04118.

**Questions:**

- in Table 2, does "No Target" simply correspond to best-of-n sampling with the draft model? If so, it would help to clarify this (if it is something else, please also clarify).
- in the experiments, the authors say they compare to beam search. However, looking at Figure 1, I think what they actually compare against is the following baseline: In each step, create $n$ drafts, select the one with the highest reward, and use that as a starting point in the next step. (This is different from beam search, since in beam search, all beams can have different histories). In particular, comparing against this baseline seems much more reasonable than comparing against beam search. Could the authors clarify this, please?
- could the authors explain why they believe they see an improvement over the "beam search with the target model" baseline? If "beam search" is what I think it means (see above), I don't see why SPECS should ever perform better than beam search with the target model. Could this be attributed to the draft model actually performing better than the target model on these datasets or on certain samples?
- the authors say they analyze the algorithm with the idealized PRM "for the purpose of keeping the theoretical presentation cleanest". Does this mean a similar result also holds with any "out-of-the-box" PRM? If so, this should certainly be included in the paper. If not, the authors should clarify this.
- it would be interesting to see how SPECS scales beyond $n=16$

Note: In the appendix, line 721: not a complete sentence (probably a period instead of a comma)

---

> ### Author Response · Authors · 2025-11-22
> **Response to Reviewer pgD8**
>
> Thank you for taking the time to review our paper, and for the valuable comments. We appreciate that you recognized our algorithm as novel and nice, and that our presentation was easy to follow.
>
> > **R3-1:** Contribution of SPECS and comparison with RSD
>
> **A.** While Reward-Guided Speculative Decoding (RSD) [1] and SPECS both utilize draft and reward models, their objectives are distinct and our algorithm also performs significantly better with equal resource budgets. **RSD is an acceleration method**: It is designed to speed up a single decoding stream. It essentially replaces the rejection sampling criterion of standard speculative decoding with a heuristic reward threshold and does not natively handle parallel exploration. **SPECS is a test-time scaling method:** Our goal is not just to be faster, but to optimize the **accuracy-latency Pareto frontier.** By generating $N$ candidate drafts in parallel and using a principled selection mechanism, we leverage the draft model to explore a much larger search space than would be possible with the target model alone within the same latency budget.
>
> We also point out that GSI [2] (ArXiv, June 2025) is to be considered a concurrent work relative to this submission. We are happy to share any additional information (such as our Arxiv identifier) necessary to substantiate this point.
>
> However, notwithstanding this fact, even considering GSI, SPECS has several novelties, such as the **Dynamic Switching** mechanism we propose (starting with the target model and switching to the draft model only for high-reward/easier steps), our stronger theoretical guarantees, and the insights in Section 2 of the paper. The former switching mechanism is crucial because, as shown in **Figure 2b** in our paper, the draft model’s performance degrades significantly on "hard" (low-reward) steps. GSI lacks this adaptive mechanism and applies the tilted reward uniformly. Indeed, this difference results in our algorithm outperforming GSI based on an additional experiment we run below in Table 5. Our results are stated in the current form as the only common datasets are MATH500 and OlympiadBench and common numbers they share are n=4,n=16. Note that we unfortunately cannot compare latency numbers because the GSI paper does not share latencies per benchmark but only shares the average latency across benchmarks (this includes some additional benchmarks we did not run). Furthermore, notice that our results presented below are slightly higher than what we have in the paper, this is because GSI uses a reasoning model “Qwen2.5-Math-Instruct”, while we use the “Qwen2.5-Instruct” model in all our experiments in the original submission. In the attached experiments below we also use the Qwen2.5-Math-Instruct reasoning model to make the comparison fair. As shown, our method achieves superior performance than GSI in all considered settings.
>
> **Table 6: SPECS (with Math-Instruct reasoning models) vs. GSI**
>
> | Dataset            | Beam Width (n) | SPECS (Acc) | GSI (Acc) |
> |:--------------------:|:----------------:|:---------------------------------------:|:----------------------------------------:|
> | MATH500      | 4              | 82.4%                             |     81.2%                     |
> | MATH500      | 16            | 85.7%                             |      82.2%                        |
> | OlympiadBench      | 4            | 41.7%                    |      39.7%                   |
> | OlympiadBench      | 16          | 44.2%                   |        40.8%                     |

---

> ### Author Response · Authors · 2025-11-22
> **Response to Reviewer pgD8 (Part 2)**
>
> ### Theoretical considerations
>
> > **R3-2:** Assuming an idealized PRM is a strong condition.
>
> **A.** The reviewer is fair to point out that our theory assumes that the PRM under consideration satisfies a strong guarantee, which is that it equals the optimal soft-advantage function. While this assumption has been used previously in the literature, only very recently (post ICLR submission deadline) have weaker versions of this assumption been considered in the literature. In particular, the only weakening of this assumption which the authors are aware of in the test-time scaling regime is the very recent work of [1] (October 2025), who assume that the PRM under consideration satisfies a pointwise error bound to the idealized PRM. Namely,
>
> $$\\| r\_{\texttt{PRM}} - \underbrace{r^{\text{target}}\_{\texttt{PRM}}}\_{\text{idealized PRM}} \\|_\infty \le \varepsilon\_{\texttt{PRM}}$$
>
> Such assumptions are compatible with our analysis. A modification of the proof in the paper to handle this setting results in an upper bound in Theorem 1 which scales with an additional term which scales as $O(\beta H \varepsilon_{\text{PRM}})$. **We have included a short discussion in Section 3.3 and the above result as Theorem 2 in the paper.**
>
> > **R3-3:** Sampling the number of beams from a Poisson distribution is unrealistic.
>
> **A.** Our main result (Theorem 1) assumes that the number of beams is drawn from a Poisson distribution. We impose this assumption for two reasons: (1) For large $n$, $\textsf{Poi} (n)$ is tightly concentrated around $n$, so the number of beams used does not vary drastically, (2) it enables making the proof cleaner. However, this assumption is strictly for convenience’s sake. To address the reviewer’s question, **we have modified all of the results in the paper to no longer rely on the Poisson sampling condition.** While in some places the resulting guarantee is slightly weaker, ($O(1/n^2)$ becomes $O(1/n^{3/2})$), this nevertheless addresses the question of the reviewer. We show that under a weaker version of the Poisson sampling assumption than what was considered in the original submission, we can strengthen our guarantees back to $O(1/n^2)$ rates.
>
> > **R3-3:** Minor comments.
>
> **A.**  In the proof of our main result, we assume that the number of reasoning blocks $H$ is fixed across all problems. This is purely for convenience - it suffices to let $H$ denote any upper bound on the ``reasoning budget’’ of the problem - i.e., the maximum number of blocks we are allowed to decode from models. It is permissible for solutions to be of varying lengths. Similarly, our results do not need to assume that individual blocklengths (number of tokens per block) are constant.
>
> [1] Rohatgi, D., Shetty, A., Saless, D., Li, Y., Moitra, A., Risteski, A., & Foster, D. J. (2025). Taming Imperfect Process Verifiers: A Sampling Perspective on Backtracking. arXiv preprint arXiv:2510.03149.

---

> > ### Author Response · Authors · 2025-11-22
> > **Response to Reviewer pgD8 (Part 3)**
> >
> > ### **R3-4** Additional questions.
> >
> > > “does "No Target" simply correspond to best-of-n sampling with the draft model?”
> >
> > In Table 2 in the paper, “No Target” corresponds to the case where we always use the draft model to generate the beams, but still use the target model’s loglikelihoods when computing the scoring function. As explained in responses **R2-1** and **R2-4** to the Reviewer Weav, the main purpose of this ablation is to show the usefulness of our proposed scoring mechanism, which is understood when we compare this “No Target” case to beam search with the small model.
> >
> > > Clarification about “beam search”:
> >
> > In the paper when we refer to beam search, we mean blockwise best-of-n (BoN), where we have N generations from a model and subselect the best among them based on a reward function to continue generations from (this is repeated iteratively in each block, where the selected beam spawns $N$ children beams). The algorithm the reviewer refers to as “beam search” has also been referred to by the same name in the literature, and we will clarify in the paper that we are indeed referring to the version above (blockwise BoN, which corresponds to setting the “beam size” to be $1$). The reason we consider beam size to be equal to $1$ is purely for latency reasons, as our implementation can make maximal usage of the prefix cache. We leave generalizing our method to large beam sizes as interesting future work.
> >
> > > “could the authors explain why they believe they see an improvement over the "beam search with the target model" baseline?”
> >
> > Thanks for asking this question. The reason why we see occasional improvements over beam search (blockwise BoN) with the target model is explained by our ability to choose the threshold value $\tau$. In particular, as observed in Fig 6b in the paper, when we threshold to those partial responses which already have very high reward (0.7) under the PRM, and continue the generation from the small model, we observe that the small model can sometimes be a more accurate “completer” than the big model. For beams which have low scores that do not meet this high threshold, we stick to the large model anyways. In effect, we see an ensembling effect where each (large/small) model is able to supplant the weaknesses of the other, as we vary the reward of the current partial trace.
> >
> > > Use idealized PRM "for the purpose of keeping the theoretical presentation cleanest". Does this mean a similar result also holds with any "out-of-the-box" PRM?
> >
> > We have changed the language in the paper to clarify this: our intention with this sentence is to say that we analyze the algorithm in the idealized setting so that the analysis captures the key features of the algorithm, without going into details of how to model the noise in the PRM, or additional assumptions on how the PRM behaves. We have clarified in **R3-2** that our analysis can tolerate misspecification / a non-ideal PRM as well, and have included some discussion in the paper as well. In general some assumption on the PRM is necessary to be able to analyze the KL divergence to the optimal policy $\pi^\star_\beta$: it is an interesting question to come up with new models of error/misspecification for the PRM: this is a question which deserves dedicated attention, and which we leave for future work.

---

### Official Review · Reviewer_Weav · 2025-10-31

**Soundness:** 2
**Presentation:** 3
**Contribution:** 2
**Rating:** 4
**Confidence:** 2

**Summary:**

This paper introduces SPECS, an inference framework to speed up reasoning for LLMs. Building on beam search and speculative decoding, SPECS uses a smaller model to draft candidate reasoning traces and a reward model to guide selection, dynamically switching to a larger model when needed. Experiments on math reasoning benchmarks show that SPECS achieves comparable or higher accuracy than beam search while reducing latency by up to 18%.

**Strengths:**

1. The paper is well-written and easy to follow.
2. The approach balanced between theory and practice, where the theoretically-rooted method also works well in practice, advancing the pareto frontier of accuracy and latency.
3. The experiments cover a wide range of models from different families.

**Weaknesses:**

1. As far as the reviewer understands, the practical difference between SPECS and RSD is that (1) SPECS integrates LLM logits for reward modeling, and (2) SPECS uses the large model in the initial step, whereas RSD uses the small model in the initial step. It remains questionable which of the two actually contributes to the accuracy gains. This is especially important, as the theory-grounded reward design is the main novelty of the paper.

2. The evaluation is limited to math benchmarks. It remains questionable if the approach will generalize to non-mathematics benchmarks, such as GPQA or coding benchmarks.

3. Some claims need to be toned down.

3-1. The authors claim that existing test-time scaling approaches usually optimize for total FLOPs, and SPECS optimizes for latency. However, many existing works on efficient test-time scaling focus on optimizing practical efficiency metrics, such as inference latency or number of generated tokens (which translates to decoding latency) [1][2]. This should be toned down throughout the paper.

3-2. Observation 1 in Section 2 is already well-known, especially given that it is the precise motivation of speculative decoding (which has already become a large field of study [3]). This needs to be toned down, as the current presentation suggests that it is a new observation made by the authors.

[1] Yang et. al., Dynamic Early Exit in Reasoning Models, preprint

[2] Yu et. al., Think Smarter not Harder: Adaptive Reasoning with Inference Aware Optimization, preprint

[3] Xia et al., Unlocking Efficiency in Large Language Model Inference:A Comprehensive Survey of Speculative Decoding, ACL 2024 Findings

**Questions:**

Will SPECS outperform RSD + using large model in the initial step? Also, will SPECS + using small model in the initial step outperform RSD? (This result will clearly present the practical effectiveness of the proposed scoring function.)

---

> ### Author Response · Authors · 2025-11-22
> **Response to Reviewer Weav**
>
> Thanks a lot for your valuable feedback. We appreciate the reviewer for acknowledging that the paper provides a theoretically-rooted method that also works well in practice.  We also appreciate that you find the paper well-written and easy to follow. Below, we address the raised weaknesses.
>
> > **R2-1:** Attributing the accuracy gains of SPECS to using LLM logits for reward modeling and using large model at the initial step.
>
> **A.** Thank you for your question, we think this touches on attributing the overall success of SPECS to 2 key components is crucial. In our ablation experiments in Table 2 (page 9) of our submitted paper, No Target experiments refer to experiments where we do not generate any beams from the target model, and generate all the beams from the draft model, but keep the SPECS scoring mechanism which uses the logits of the large model. When we switch the scoring mechanism back to only using PRM rewards, this goes back to beam search with the small model (**BS(S)**). Thus, comparing these two columns (No Target vs. BS(S)) in Table 1, gives us insights into the role of the scoring mechanism - we observe that No Target uniformly outperforms **BS(S)**, showing the utility of our scoring mechanism with target model logits. On top of these positive results with just using the SPECS scoring mechanism, our proposed dynamic switching mechanism which begins generations from the target model beam and switches to generating from the draft model further improves the performance of SPECS to nearly match the beam search with the big (target) model **BS(B)** on most benchmarks while having lower latency numbers.
>
> Furthermore, to address the reviewer’s concerns, we carry out 3 additional experiments to clarify this issue. First, in the RSD algorithm, we modify it by using the large model in the initial step and the draft model after the usage of the target model. The results indicate that SPECS trumps this algorithm in all cases:
>
> **Table 3: SPECS vs. RSD with initial draft generated from large model**
> | Dataset            | Beam Width (n) | SPECS (Acc / Latency) | RSD with big model first (Acc / Latency ms) |
> |:--------------------:|:----------------:|:---------------------------------------:|:----------------------------------------:|
> | AMC              | 4              | 59.2% / 14.8s                             |    43.3% /  16.8s              |
> | AMC              | 8              | 62.5% / 20.7s                             |    39.2% / 21.2s                         |
> | AMC              | 16            | 68.3% / 28.8s                             |    48.3% / 26.2s                         |
> | MATH500      | 4              | 77.0% / 9.83s                             |     70.0% /   13.8s                      |
> | MATH500      | 8              | 79.5% / 11.3s                             |     66.7% / 17.8s                        |
> | MATH500      | 16            | 81.0% / 15.8s                             |      71.3% / 22.1s                        |
> | OlympiadBench      | 4            | 41.3% / 13.1s                      |      28.0% / 23.3s                       |
> | OlympiadBench      | 8            | 41.7% / 16.9s                      |      32.7% / 26.0s                      |
> | OlympiadBench      | 16          | 44.0% / 24.6s                      |        30.7% / 35.4s                     |

---

> ### Author Response · Authors · 2025-11-22
> **Response to Reviewer Weav (Part 2)**
>
> Next, we compare SPECS with another variant where we do not use large model likelihoods in the scores, and do regular beam search once we choose to sample the beams from the draft model. This experiment complements the aforementioned two experiments and the positive results demonstrate the benefit of using our score function:
>
> **Table 4: SPECS vs. BS(B) + dynamic switching to BS(S) (i.e., SPECS with no logits in score)**
> | Dataset            | Beam Width (n) | SPECS (Acc / Latency) | SPECS with no logits (Acc / Latency ms) |
> |:--------------------:|:----------------:|:---------------------------------------:|:----------------------------------------:|
> | AMC              | 4              | 59.2% / 14.8s                             |    43.2% / 12.5s            |
> | AMC              | 8              | 62.5% / 20.7s                             |    47.5% / 17.9s                       |
> | AMC              | 16            | 68.3% / 28.8s                             |    48.6% / 28.1s                         |
> | MATH500      | 4              | 77.0% / 9.83s                             |     71.5% / 8.73s                  |
> | MATH500      | 8              | 79.5% / 11.3s                             |     73.3% / 10.6s                 |
> | MATH500      | 16            | 81.0% / 15.8s                             |     74.5% / 14.5s                     |
> | OlympiadBench      | 4            | 41.3% / 13.1s                      |     32.5% /16.6s                    |
> | OlympiadBench      | 8            | 41.7% / 16.9s                      |      32.8% / 23.8s                   |
> | OlympiadBench      | 16          | 44.0% / 24.6s                      |      33.1% / 27.1s                  |
>
> Finally, we run an experiment where draft generation in SPECS is initiated with the small model. Note that in this case, we achieve very similar accuracy numbers to the original SPECS we proposed, but starting with the small model appears to incur a latency overhead. The reason for this is stated in **Observation 2** in the paper: while the small model is very good at successfully completing steps after the few steps of a large model, we notice that in the first few steps, a threshold value of $\tau \in [0.7,0.9]$ tends lead to most draft model outputs to be rejected in the initial rounds. In the steps where all drafts are rejected, drafts are later generated from the target model, but we unnecessarily incur a latency overhead of having decoded from the draft model. This gives a clean motivation for why we decided to begin with generating from the target model in SPECS. We will make this rationale clearer in the paper and provide relevant insights accordingly. These numbers are also better than the RSD numbers we indicate in the paper, which starts with the small model at the initial step as the reviewer indicated, showing the usefulness of our scoring mechanism.
>
> **Table 5: Comparison of SPECS with initial draft generation with small model**
> | Dataset         | Beam Width (n) | SPECS (Acc / Latency s) | SPECS (start w/ small model) (Acc / Latency s) | RSD (Acc / Latency s) |
> | :-------------- | :------------: | :----------------------: | :--------------------------------------------: | :-------------------: |
> | **AMC**         | 4              | 59.2% / 14.8s            | 58.9% / 18.5s                                  | 31.2% / 12.6s         |
> |                 | 8              | 62.5% / 20.7s            | 61.4% / 21.9s                                  | 49.2% / 20.3s         |
> |                 | 16             | 68.3% / 28.8s            | 67.9% / 29.2s                                  | 50.0% / 25.0s         |
> | **MATH500**     | 4              | 77.0% / 9.83s            | 77.5% / 13.0s                                  | 77.8% / 10.1s         |
> |                 | 8              | 79.5% / 11.3s            | 78.9% / 12.7s                                  | 78.3% / 11.8s         |
> |                 | 16             | 81.0% / 15.8s            | 81.0% / 17.2s                                  | 79.0% / 16.3s         |
> | **OlympiadBench** | 4            | 41.3% / 13.1s            | 41.0% / 15.3s                                  | 33.0% / 17.3s         |
> |                 | 8              | 41.7% / 16.9s            | 41.3% / 18.1s                                  | 33.3% / 21.7s         |
> |                 | 16             | 44.0% / 24.6s            | 44.2% / 27.8s                                  | 33.5% / 28.7s         |

---

> ### Author Response · Authors · 2025-11-22
> **Response to Reviewer Weav (Part 3)**
>
> > **R2-2:**  Evaluations being only limited to math benchmarks
>
> **A.** We thank the reviewer for these suggestions. We are working on adding additional experiments on commonsense and general-purpose reasoning datasets, and will share the results in a few days. We decided to post the current version of the rebuttal addressing your other concerns to be able to get your feedback on our response early on.
>
> > **R2-3:** Some claims need to be toned down
>
> **A.** Thank you for your suggestions. We have rewritten and toned down the claims related to Observation 1 in the paper: we have clarified that our purpose of stating this observation is mostly to be able to demonstrate the actual latency gaps present when implemented with current SOTA open-source reasoning and PRM models - one should not expect that gaps between small and large models persist. This crucial experiment is what we use to inform the design of our algorithm that achieves the pareto-frontier of the latency-accuracy tradeoff. We also clarify for the purpose of the rebuttal that our proposed algorithm is orthogonal to the references [1] and [2] attached in the reviewer’s response. In a nutshell, the aim of these approaches is to, respectively, cut down the total number of tokens decoded, and to allocate more tokens to hard problems and fewer tokens to easier problems. In our revised submission, we have tried to make it clear that we are not entirely putting forward the problem of optimizing the latency-accuracy tradeoff ourselves, but designing a new algorithm to tackle this problem within the context of test-time scaling methods, and designing a theoretical model (which have not been considered in prior work) and empirical analyses surrounding this tradeoff.
>
> > **R2-4:** Questions about comparing SPECS to RSD (with large model in initial step) and SPECS (with small model in initial step) to RSD
>
> **A.** Thank you for asking this question that helps clarify the significance of the proposed scoring function. As we explained in **R2-1** above, in our ablation studies in Table 2, by comparing the version of SPECS with all beams generated by the small model to beam search with small model as we have in Table 2 of the paper and conducting **three additional experiments** we provided, we hope that we clarified the importance of the proposed scoring mechanism. We are happy to answer any questions the reviewer further has.
>
>
> We thank the reviewers for the nice questions and suggestions. In light of the new experiments and explanations, we hope that we have adequately addressed most concerns. We are happy to address any more questions, and if the reviewer is content with our responses, we would appreciate any reconsideration to the score of our paper.

---

> ### Author Response · Authors · 2025-11-27
> **New Experiments on GPQA Dataset**
>
> Dear Reviewer Weav,
>
> Thank you for your comments once more. As we promised, we ran more results on the GPQA dataset using the Qwen family of models (Qwen2.5-1.5B-Instruct and Qwen2.5-7B-Instruct)  and the same Qwen2.5-7B based PRM we had in our paper. The results are presented below. Note that SPECS achieves much better latency compared to the Beam Search with Large Model while achieving similar accuracy results, demonstrating the applicability of SPECS to reasoning problems other than mathematics:
>
> | Dataset | Beam Width (n) | SPECS (Acc / Latency) | Beam Search (Big Model) (Acc / Latency) | Beam Search (Small Model) (Acc / Latency) |
> |:-------:|:--------------:|:-----------------------:|:-------------------------------------------:|:--------------------------------------------:|
> | GPQA | 4  | 30.8% / 9.00s  | 34.2% / 18.6s | 17.8% / 7.77s |
> | GPQA | 8  | 32.2% / 21.83s | 32.5% / 23.1s | 22.6% / 10.31s |
> | GPQA | 16 | 34.5% / 22.42s | 33.3% / 28.3s | 24.2% / 15.08s |
>
>
> We hope that with the new experiments and explanations, we have adequately addressed most concerns and are happy to answer more questions if any; and we would appreciate it if the reviewer could reconsider the score of our paper.
>
> Best regards,
>
> Authors

---

### Official Review · Reviewer_dw1G · 2025-10-31

**Soundness:** 2
**Presentation:** 2
**Contribution:** 2
**Rating:** 4
**Confidence:** 3

**Summary:**

This paper proposes SPECS, improving the latency-accuracy trade-off for test-time scaling in LLMs, particularly for reasoning tasks. It starts by generating from the target model and dynamically switches to speculative drafting for high-reward (from a Process Reward Model) traces to minimize accuracy drop. The two main components are a *soft verification* scoring function that combines the PRM score with a log-density ratio, and a *dynamic switching* mechanism that starts with the target model and only switches to the draft model once the PRM score exceeds a certain threshold. The authors claim this method matches or exceeds the accuracy of a large-model beam search while reducing latency by up to ~18% on math reasoning benchmarks.

**Strengths:**

1. Unlike traditional speculative decoding, SPECS starts with the more capable target model. It features a novel dynamic switching mechanism that transitions to the faster draft model only after a high-reward reasoning trace has been identified.
2. The insight that a draft model is sufficient for easier steps (identified by a high PRM score) and that the system should start with the target model is well-motivated.
3. The authors provide a clear theoretical motivation for their method.

**Weaknesses:**

1. As shown in Table 3, the hyperparameters β (inverse-temperature for the soft verification) and τ ( dynamic switching threshold) are tuned for each specific dataset and beam width. This raises questions about the method's sensitivity to the choice of the parameters and the practical overhead of deploying it, as it would seem to require a new, exhaustive hyperparameter search for any new task. The paper provides no robustness analysis to these choices, justification for the specific numbers, or a simple heuristic for setting these parameters (why MATH500 and OlympiadBench use 100 questions and AMC23 uses the whole 40 questions). Furthermore, the tuning methodology itself is inconsistent: validation sets are small and inconsistent (100 questions for two datasets, but only 40 for AMC23), and selecting parameters that give the "best accuracy vs. latency" seems a flexible metric.  In addition, the authors state that "When we condition on the traces having high reward (as computed by a PRM) at the 8th step, we observe that completing the trace using the draft model affects accuracy very minimally compared to completing with the target model", which implies that the decision to switch seems to be example-dependent. Will a different example have a different dynamic switching threshold?

2. The entire dynamic switching mechanism depends on a reliable PRM. The paper uses a strong, existing PRM, but will the performance of SPECS degrade if the PRM is of lower quality, e.g. PRM confidently assigns a high score to an incorrect "easy" path? In addition, will using a different PRM fundamentally change the accuracy vs. latency trade-offs reported in the paper?

3. The method is only evaluated on mathematical reasoning, a highly structured domain where "correctness" is well-defined and PRMs are known to be effective. It remains unclear how SPECS would perform on other types of reasoning tasks (commonsense reasoning tasks), open-ended generation tasks (e.g., summarization), or other domains like code generation.

4. There are also concerns about the presentation of the results in Table 1. The authors only bold the SPECS accuracy score even when the accuracy is the same as the baselines. For example, on AMC23 (n=4), BS(B) achieves 59.2±1.2 and SPECS achieves 59.2±3.1, but only the SPECS result is bolded. This can be misleading. Furthermore, many accuracy improvements are marginal. Are these improvements statistically significant? In addition, the authors selectively bold only accuracy, while latency and latency/step metrics (where SPECS is not always the best performer) are left unbolded.

**Questions:**

See in weaknesses.

---

> ### Author Response · Authors · 2025-11-22
> **Response to Reviewer dw1G**
>
> Thank you for your valuable comments. We appreciate that you find our method well-motivated and the theoretical analysis clear. We address your questions and comments in detail below:
>
> > **R1-1:** What is the justification for the current parameters (eg. dataset sizes, etc). How sensitive is the method to hyperparameter choice? Would it require a new, exhaustive hyperparameter search for a new task?
>
> **A.** Current Parameters Selection: For MATH500 and OlympiadBench, we select a subset of examples in the dataset as a hold-out validation set to perform hyperparameter search  for best accuracy-latency tradeoff. The size of the dataset subset is chosen to be 100 for efficiency. For AMC2023, we used the full dataset as validation and test dataset since it only has 40 questions.
>
> **Hyperparameter Sensitivity:**
> We appreciate the reviewer’s suggestion to analyze sensitivity to hyperparameters. We refer the reviewer to point 1 of our general response. In summary, our method has two hyperparameters, the inverse temperature $\beta$ and switching threshold $\tau$. In particular, the existence of these hyperparameters gives the developer some flexibility to tune them based on the usecase, for example choosing a higher threshold $\tau$ gives more accurate but slower responses in general compared to a lower $\tau$, and choosing a high $\beta$ means we rely on the PRM rewards more and if the PRM is thought to be low quality or noisy, the developer can choose a lower $\beta$ value.
> To address the reviewer's concern, we run some additional ablations across all datasets considered in the paper to demonstrate the robustness of our method to hyperparameters. We see that selecting **$\beta = 2 \times L$** where $L$ equals the maximum token budget per step is a stable choice to achieve desirable accuracy-latency tradeoff. With this choice of $\beta$, the search space for $\tau$ can be restricted to a small subset $\\{ 0.7, 0.8, 0.9 \\}$.  We report the accuracy and latency numbers of restricted search compared to an exhaustive search over $\beta$ and $\tau$
>
> - **Constrained search:**
> $\beta \gets 2 \times L$, where $L$ is the maximum per-step decoded token length,
> $\tau \in \\{ 0.7, 0.8, 0.9\\}$.
>
> **Table 1: Highly constrained hyperparameter search does not result in much drop in accuracy compared to an exhaustive search on many datasets**
> | Dataset            | Beam Width (n) | Exhaustive Search (Acc / Latency) | Constraint Search (Acc / Latency ms) |
> |:--------------------:|:----------------:|:---------------------------------------:|:----------------------------------------:|
> | AMC              | 4              | 59.2% / 14.8s                             |    53.1% /  13.3s              |
> | AMC              | 8              | 62.5% / 20.7s                             |    58.4% / 20.2s                         |
> | AMC              | 16            | 68.3% / 28.8s                             |    64.8% / 26.4s                         |
> | MATH500      | 4              | 77.0% / 9.83s                             |     77.0% /   9.8s                      |
> | MATH500      | 8              | 79.5% / 11.3s                             |     75.3% / 11.0s                        |
> | MATH500      | 16            | 81.0% / 15.8s                             |      81.0% / 15.8s                        |
> | OlympiadBench      | 4            | 41.3% / 13.1s                      |      38.5% / 12.5s                       |
> | OlympiadBench      | 8            | 41.7% / 16.9s                      |      41.0% / 18.1s                      |
> | OlympiadBench      | 16          | 44.0% / 24.6s                      |        42.6% / 23.8s                     |
>
>
> Even with this highly restricted tuning (only $\tau$ over a narrow range), performance remains similar to the exhaustive setting. This demonstrates that our method is not overly sensitive to hyperparameters. We will add more detailed analysis in the final version of the paper.
>
> As mentioned in the point 1 of the general response, the ability to choose different values of $\tau$ is an important flexibility and feature of SPECS: use-cases which necessitate faster execution can accommodated by choosing lower values of $\tau$, and use-cases which necessitate higher accuracy can be accommodated by choosing higher values of $\tau$. Thus, the key hyperparameter to be tuned in our method is the inverse-temperature $\beta$, which we observe low sensitivity toward (cf. Table 1 attached above).

---

> ### Author Response · Authors · 2025-11-22
> **Response to Reviewer dw1G (Part 2)**
>
> > **R1-2:**  The authors state that "When we condition on the traces having high reward (as computed by a PRM) at the 8th step, we observe that completing the trace using the draft model affects accuracy very minimally compared to completing with the target model", which implies that the decision to switch seems to be example-dependent. Can a different example have a different dynamic switching threshold?
>
> **A.** We would like to clarify that analysis in Figure 2b and Appendix D5 is not done for a single example, but on a subset of the Math500 dataset (100 problems) where we used a fixed PRM threshold ($0.5$) for all the trajectories. We have clarified this confusion in the revision of the paper. Taking a step back, in the SPECS algorithm the switching threshold $\tau$ can be searched over using a lightweight hyperparameter search as we do, to be adaptive to the domain/task/dataset. We agree with the reviewer that the ability to choose a different value of $\tau$ on a per-example basis may enable the performance of SPECS to improve further, and the statistics of PRM scores across previous steps may be useful in this regard. We argue that this direction deserves a dedicated study of its own, and we leave it for future work.
>
>
> > **R1-3:** Will the performance of SPECS degrade if the PRM is of lower quality?
>
> A: We agree with the reviewer that dynamic switching to an extent depends on a reliable PRM, but the reliance on an imperfect PRM can be viewed as a deficiency of any work on RM-based test-time scaling. However, as discussed in point 2 of our general response, the fact that SPECS combines log-likelihood ratios with PRM scores provides robustness and regularization to noisy and imperfect reward models. In particular, the candidate selection step (The subsample subroutine) can be shown to be robust to the imperfections in the PRM because it also makes use of the log-probs of the large model as a decision-making signal (in addition to the PRM).
>
>
>
>
> To demonstrate this, we run some additional ablations to check the degradation in performance when SPECS and beam search are carried out with a weaker PRM. We carry out an experiment in this regard:
>
> **Adding controlled noise to the PRM (Gaussian noise)**: We ablate the quality of the PRM by adding explicit Gaussian noise to the outputs of the model. In the particular, the PRM we consider is $\tilde{r}\_{\texttt{PRM}} = r_{\texttt{PRM}} + Z$, where $Z$ is independent $0$-mean Gaussian noise with standard deviation $0.1$. In this regime, we see that the performance gap between SPECS and beam search with small model increases significantly, while the performance of SPECS still remains competitive with beam search with the large model. While we do not carry out any additional hyperparameter search prior to running SPECS in this setting. The results are presented in Table 2 below.
>
> **Table 2: Evaluation of SPECS with a PRM with added Gaussian noise**
> | Dataset         | Beam Width (n) | Beam (Qwen 1.5B) (Acc / Latency s) | Beam (Qwen 7B) (Acc / Latency s) | SPECS (Acc / Latency s)      |
> |:---------------:|:--------------:|:-----------------------------------:|:--------------------------------:|:----------------------------:|
> | AMC             | 4              | 36.25% / 9.55s                      | 43.33% / 14.44s                  | 46.67% / 16.50s              |
> | AMC             | 8              | 34.38% / 12.07s                     | 50.83% / 20.59s                  | 50.83% / 19.64s              |
> | AMC             | 16             | 43.13% / 19.73s                     | 46.67% / 26.62s                  | 49.17% / 27.57s              |
> | MATH500         | 4              | 53.33% / 7.07s                      | 76.67% / 11.18s                  | 78.00% / 11.07s              |
> | MATH500         | 8              | 62.00% / 9.77s                      | 78.67% / 14.21s                  | 71.33% / 12.49s              |
> | MATH500         | 16             | 59.33% / 15.32s                     | 73.33% / 18.57s                  | 72.67% / 18.76s              |
> | OlympiadBench   | 4              | 26.00% / 11.38s                     | 30.67% / 14.38s                  | 30.67% / 14.10s              |
> | OlympiadBench   | 8              | 25.00% / 15.83s                     | 34.00% / 20.29s                  | 35.33% / 19.99s              |
> | OlympiadBench   | 16             | 29.33% / 25.38s                     | 32.67% / 25.49s                  | 37.00% / 27.99s              |

---

> > ### Author Response · Authors · 2025-11-22
> > **Response to Reviewer dw1G (Part 3)**
> >
> > > **R1-5**: It remains unclear how SPECS would perform on other types of reasoning tasks (commonsense reasoning tasks), open-ended generation tasks
> >
> > **A.** We thank the reviewer for the suggestion on validation of our method on tasks beyond mathematical reasoning. We are currently running some additional evaluations on commonsense reasoning datasets and will share the results soon. We decided to post our rebuttal addressing your other concerns, to get any additional feedback early on.
> >
> > > **R1-6:** Concerns about the presentation of the results in Table 1.
> >
> > **A.**  Thanks for raising this crucial point about the paper. As you indicated, the latency improvements is the main contribution and aim of the paper. The reason why we highlight SPECS when both SPECS and BS(B) achieve 59.2 for AMC23 (n=4) is that we achieve the same results with lower latency numbers. Note that highlighting the smaller latency numbers in Table 1 in the paper for all the methods would not be fair, because a visibly inferior algorithm, beam search with the small model BS(S), achieves faster results than any other method in all cases (therefore boldening the latency is not really insightful). We agree with the point that the main takeaway of these experiments is that SPECS does not necessarily aim to achieve higher accuracy, but aims to be faster than beam search with the big model while achieving on-par accuracy. We will clarify the emphasis in the paper, and are open to any suggestions to presenting this more clearly.
> >
> > We thank the reviewer for your time and constructive suggestions on the paper. We hope that we have addressed the concerns raised with the new experiments (aside from those pertaining to experiments which are still underway) and discussion. We would also appreciate it if the reviewer would reconsider their score to our paper.

---

### Author Response · Authors · 2025-11-22
**General Response**

**To address the major comments raised by reviewers, we have made the following revisions to the paper (each of which is discussed in the sequel):**

1. **Hyperparameter Sensitivity (Table 1):** We show that SPECS demonstrates robust performance across benchmarks given fixed hyperparameters throughout. We empirically demonstrate that only a small array (3 values) of hyperparameter values can generalize and be used across different settings.

2. **Ablating PRM quality:** We ablate the robustness and performance of SPECS against the quality of the PRM by using it with a noisy PRM and showing that it is still on the pareto frontier of latency-accuracy tradeoff.

3. **Non-math reasoning benchmarks:** We are currently in the process of evaluating the performance of SPECS on general purpose /commonsense reasoning benchmarks, and will report results for these as soon as possible.

4. **Theoretical results:** We include two new theoretical results to address the concerns raised, (1) In the newly added Section 3.3 of the paper, we show that SPECS degrades gracefully with misspecification error when the PRM is not ideal, and (2) We also modify our proof of Theorem 1 to show that the Poisson sampling assumption can effectively be eliminated from all of the results in the paper. This further demonstrates that our proposed method is theoretically grounded, which could be of independent interest.

&nbsp;
## **1.** Hyperparameter Selection
The SPECS algorithm has two hyperparameters, the inverse temperature $\beta$ and switching threshold $\tau$. Empirically, across all datasets considered, we see that selecting **$\beta = 2 \times L$** where $L$ is the maximum per-step token budget is a stable choice of this hyperparameter. On the other hand, **the $\tau$ hyperparameter should be viewed as a knob which can be varied to move along the latency-accuracy pareto curve.** Selecting this hyperparameter should be viewed as a function of the precise latency requirement, which may vary from setting to setting. In Figs. 3 and Table 1 in the paper, we chose to select $\tau$ in a manner which minimizes the latency while the drop in accuracy compared to beam search with the large model is negligible (within one standard error). **In the attached Table 1, we plot how the accuracy and latency vary as a function of $\tau$, holding $\beta = 2 \times L$.** We see two things,

- Selecting $\tau \approx 0.7$ always appears to be a good value of this hyperparameter and results in minimal performance degradation compared to beam search with the big model while usually reducing latency.

- Varying $\tau$ in the range $\\{ 0.7,0.8,0.9 \\}$ results in a range which captures most of the working range of the latency-accuracy pareto curve. This range does not appear to depend on the dataset.

Having said this, even ignoring the above considerations, we point out that the methodology of tuning hyperparameters on a validation set (when available) and on the evaluation set (when the size of the dataset is very small), is standard practice in the literature on test-time scaling. Indeed, in most cases, the latency spent in optimizing hyperparameters of test-time scaling methods is amortized (and in fact dominated by many orders of magnitude) by the downstream savings in inference latency, which justifies such post-deployment tuning. While we do not observe much sensitivity to hyperparameters in our experiments, it is entirely possible that the best choice of algorithm hyperparameters, more broadly, is domain sensitive: this will necessitate some level of tuning in practice.

---

> ### Author Response · Authors · 2025-11-22
> **General Response (Part 2)**
>
> ## **2.** Does the quality of the PRM matter?
>
> PRMs have become a standardized tool in domains like math reasoning, both for inference-time optimization as well as for post-training. However, while these models offer dense process feedback, they can be noisy. **In the attached Table. 1, we ablate the performance of our algorithm PRM quality in a controlled manner by adding external Gaussian noise to the PRM.
>
> We observe that even with moderate noise variance, the performance of SPECS does not drop significantly, showing that it is robust to noise in the PRM. We select the $\tau$ hyperparameter in this case so that the latency is close to that of beam search with the big model and see that **SPECS is resilient to noise in the PRM**. This can be attributed partly to the fact that the effective "reward function" we use is $\beta r_{\texttt{PRM}} (y|x) + \log ( \pi_{\text{target}} (y|x) / \pi_{\text{draft}} (y|x) )$. Thus, even though the PRM $r_{\texttt{PRM}}$ itself may be noisy, SPECS can mitigate the effect of this via the secondary "reward", $\log ( \pi_{\text{target}} (y|x) / \pi_{\text{draft}} (y|x) )$ which gives a useful signal to judging the quality of a response.
>
> &nbsp;
> ## **3.** What happens outside of math benchmarks?
> The main objective of this paper is to study how best to carry out test-time scaling when optimizing for the latency-accuracy pareto frontier. Consequently the majority of our analysis is for settings like reasoning where there is a benefit from process supervision and scaling compute. We are currently in the process of evaluating the performance of SPECS on additional general purpose and commonsense reasoning benchmarks, and will report these numbers as soon as possible. We decided to share our rebuttal sooner, since we believe most of the other concerns have been addressed, and look forward to seeing any additional feedback/comments the reviewers may have in the meantime.
>
> &nbsp;
> ## **3.** Assumptions in theory are strong
>
> **3a. Theoretical results require an idealized PRM**
>
> The analysis in the paper assumes that the PRM under consideration is idealized, and equals the optimal soft-advantage function. While some assumption of this nature is necessary for any test-time scaling method to collect high value under the KL regularized RL objective, we can weaken the assumption in the paper to be more realistic. **We include a new Section 3.3 of the paper, we show that if the PRM is inexact, the guarantees of SPECS degrades gracefully with the misspecification error of the PRM.** Namely, suppose the PRM satisfies the condition,
> $$\\| r\_{\text{PRM}} - r^{\text{target}}\_{\texttt{PRM}} \\|\_\infty \le \varepsilon\_{\texttt{PRM}}$$
> where recall that $r^{\text{target}}\_{\texttt{PRM}}$ is the idealized PRM, and the misspecification error in the PRM is denoted by $\varepsilon_{\texttt{PRM}}$. Under this assumption, we show that the KL divergence of the policy induced by SPECS to the optimal KL regularized policy degrades with an additional additive error term that scales as $\beta H \varepsilon\_{\text{PRM}}$.
>
> **3b. Theoretical results assume the Poisson sampling framework**
>
> The analysis in the paper assumes that the number of beams drawn in SPECS is not fixed and varies as a Poisson random variable with mean $n$: while we argue this assumption, in general, is very mild, as a Poisson random variable concentrates strongly around its mean ($=n$), **we modify the proofs of our main results Theorem 1 to remove this assumption entirely.** The new guarantees apply even when the beam-width is fixed, agreeing with the way the algorithm is deployed in our experiments.

---

### Author Response · Authors · 2025-11-27
**New Experiments on the GPQA Benchmark**

Dear Reviewers,

Once more thank you all for your constructive comments that have enhanced the depth of analysis and comprehensiveness of experiments in the paper.

We have included some more results about GPQA experiments in our response to Reviewer Weav. We are also adding this table of new GPQA experiments below. In these experiments, we see that SPECS achieves much better latency compared to the Beam Search with Large Model while achieving similar accuracy results, demonstrating the applicability of SPECS to reasoning problems other than mathematics. We also do not tune any of our hyperparameters here, and directly use the ones we reported for the MATH-500 dataset, showing the robustness of those hyperparameters.

We also uploaded the new version of our paper, with the new changes marked in blue.

We are happy to address more comments or questions if you have any, or appreciate your further feedback and reconsideration of our paper.

**Table: SPECS Results on GPQA Benchmark**
| Dataset | Beam Width (n) | SPECS (Acc / Latency) | Beam Search (Big Model) (Acc / Latency) | Beam Search (Small Model) (Acc / Latency) |
|:-------:|:--------------:|:-----------------------:|:-------------------------------------------:|:--------------------------------------------:|
| GPQA | 4  | 30.8% / 9.00s  | 34.2% / 18.6s | 17.8% / 7.77s |
| GPQA | 8  | 32.2% / 21.83s | 32.5% / 23.1s | 22.6% / 10.31s |
| GPQA | 16 | 34.5% / 22.42s | 33.3% / 28.3s | 24.2% / 15.08s |

---

### Meta-Review · Area_Chair_S1uk · 2026-01-01

**Summary:**

This paper proposes SPECS, a latency-aware test-time scaling method for LLM reasoning. SPECS builds on blockwise best-of-n / “beam search”-style iterative generation (as later clarified by the authors), and uses a small draft model to propose candidate continuations that are evaluated using a target (reasoning) model and a process reward model (PRM). The submission highlights (and reviewers identify) two main algorithmic pieces: (i) a soft verification scoring function that combines PRM reward with a log-density ratio / target-model likelihood signal, and (ii) a dynamic switching mechanism that, per reviews and rebuttal, starts with the target model and switches to the draft model when the partial trace is deemed “easy/high-reward,” aiming to reduce latency while maintaining accuracy. The paper reports matching or surpassing blockwise BoN/beam search accuracy while reducing latency (e.g., “up to 18%” in the abstract).

All three reviewers give rating = 4 (marginally below acceptance). The shared decision-driving concerns are: (1) practical hyperparameter tuning sensitivity / deployment overhead (β, τ) and inconsistent validation sizing, (2) reliance on a strong PRM and uncertainty under PRM errors, (3) limited evaluation scope (initially math-only), (4) novelty positioning relative to closely related methods (RSD, GSI), and (5) a theory–practice gap due to strong assumptions and baseline-definition ambiguity (“beam search”).

The rebuttal includes new experiments and clarifications (hyperparameter heuristics and constrained search; PRM-noise ablations; additional comparisons/ablations disentangling scoring vs switching vs “start with big model”; GPQA results; and theory adjustments removing the Poisson assumption and adding a misspecification discussion). These address several concrete reviewer questions. The rebuttal strengthens the submission and addresses multiple concrete questions raised in the reviews; however, in the discussion record, reviewers do not explicitly indicate score changes, and several concerns remain only partially resolved.

**Reviewer Concerns:**

### Addressed by the rebuttal (substantially or partially)

- Hyperparameter sensitivity / tuning protocol (Reviewer dw1G): Authors explain their validation choices: 100 held-out examples for MATH500 and OlympiadBench, and using the full 40 examples for AMC2023 because the dataset is small. Authors provide additional experiments proposing a heuristic for β (set relative to max per-step token budget) and restricting τ to a small range, plus a table comparing “exhaustive search” vs “constrained search.” Overall, this concern is partially addressed (guidance and evidence added), but the constrained-search table also shows noticeable drops in some settings (as reported by authors), so the original deployment-overhead concern is not fully eliminated.

- Whether dynamic switching is example-dependent (Reviewer dw1G): Authors clarify that the analysis cited (Figure 2b / Appendix D5) is conducted on a subset of 100 Math500 problems using a fixed PRM threshold across trajectories, not a single example; they note per-example τ adaptation is left for future work. This concern has been addressed (clarifies misunderstanding; acknowledges limitation).

- Robustness to PRM quality (Reviewer dw1G):  Authors provide a controlled ablation by adding Gaussian noise to PRM outputs and report accuracy/latency comparisons vs beam search with small vs large models under this noisy PRM. This concern is partially addressed (evidence added; also shows some degradation in certain settings per their table).

- Attributing gains: scoring function vs “start with big model” vs switching; relation to RSD (Reviewer Weav). Authors cite existing ablations (“No Target” variant vs BS(S)) and add new experiments: (1) SPECS vs RSD with big model first (their Table 3). (2) SPECS vs “SPECS with no logits in score” (their Table 4), to isolate the value of including target-model likelihood in the score. (3) SPECS starting with the small model (their Table 5), where they report similar accuracy but higher latency, and use this to motivate starting with the target model. This concern is addressed in the sense that they add targeted ablations directly responding to the question.

- Beyond-math evaluation (Reviewers dw1G, Weav): Authors add GPQA experiments (beam widths 4/8/16) with accuracy/latency comparisons to beam search with the big vs small model, and state they reuse MATH-500 hyperparameters without tuning. This concern is partially addressed (adds a non-math reasoning benchmark, GPQA, with beam widths 4/8/16 and latency/accuracy comparisons; and the authors state they reuse MATH-500 hyperparameters without tuning), though the original reviewer concern also mentioned broader domains (e.g., open-ended generation / code), which remain unevaluated in the provided discussion.

- Theory assumptions and applicability (Reviewer pgD8): Authors state they add a new section analyzing degradation under PRM misspecification error and modify proofs to remove the Poisson beam-count assumption (with some rate changes). This question is addressed at the level of stated revisions and added theoretical results.

- Clarification of what baseline is called “beam search” (Reviewer pgD8): Authors explicitly clarify that their “beam search” baseline is blockwise best-of-n (BoN): generate n candidates each block, select the best by reward, and iterate (beam size effectively set to n for latency/prefix-cache reasons). This is addressed (terminology clarified).

- Concurrency / timeline vis-à-vis GSI (private message to chairs): Authors provide arXiv identifiers and dates (GSI: June 4, 2025; SPECS: June 15, 2025) and argue additional novelty from dynamic switching. This is addressed insofar as concurrency timestamps are provided (to chairs).

### Still outstanding (based strictly on what is in the reviews + discussion)

- Novelty positioning relative to RSD / GSI (Reviewer pgD8; also related to Weav’s concern): Reviewer pgD8 characterizes SPECS’s novelty as limited, with major overlap with RSD (draft+reward-based speculative) and GSI (tilted reward / scoring), and identifies the main novelty as starting with the target model + dynamic switching. The rebuttal argues different objectives (Pareto frontier vs single-stream acceleration) and provides more experiments, but the review record still reflects that novelty is a central concern rather than a resolved point.

- Generality beyond currently evaluated tasks: While GPQA was added (providing evidence beyond math reasoning), reviewers originally questioned behavior on broader domains (dw1G mentioned commonsense/open-ended generation and other domains such as summarization/code generation). No additional results for those broader domains appear in the discussion beyond GPQA.

- Practical deployment overhead of tuning remains a live concern (dw1G): Even with heuristics and constrained search, the authors’ own constrained-search table reports noticeable drops in some settings. This does not fully resolve the reviewer’s concern about whether SPECS requires significant re-tuning on new tasks to achieve strong Pareto trade-offs.

**Reviewer Scores:**

- Reviewer dw1G (original score: 4): No explicit statement about changing score.

- Reviewer Weav (original score: 4): No explicit statement about changing score.

- Reviewer pgD8 (original score: 4): No explicit statement about changing score.

---

### Decision · Program_Chairs · 2026-01-26

Reject